# Oblique geographic coordinates as covariates for digital soil mapping

Anders Bjørn Møller[1], Amélie Marie Beucher[1], Nastaran Pouladi[1], Mogens Humlekrog Greve[1]

[1]Department of Agroecology, Aarhus University, Tjele, 8830, Denmark

*Correspondence to*: Anders Bjørn Møller (anbm@agro.au.dk)

**Abstract.** Decision tree algorithms such as Random Forest have become a widely adapted method for mapping soil properties in geographic space. However, implementing explicit spatial trends into these algorithms has proven problematic. Using x- and y-coordinates as covariates gives orthogonal artefacts in the maps, and alternative methods using distances as covariates can be inflexible and difficult to interpret. We propose instead the use of coordinates along several axes tilted at

oblique angles to provide an easily interpretable method for obtaining a realistic prediction surface. We test the method on four spatial datasets and compare it to similar methods. The results show that the method provides accuracies better than or on par with the most reliable alternative methods, namely kriging and distance-based covariates. Furthermore, the proposed method is highly flexible, scalable and easily interpretable. This makes it a promising tool for mapping soil properties with complex spatial variation.

## 1 Introduction

Machine learning has become a frequently applied means for mapping soil properties in geographic space. The most common approach is to train models from soil observations and covariates in the form of geographic data layers. The models can often provide reliable predictions of soil properties. Many researchers have used decision tree algorithms, as they are computationally efficient, do not rely on assumptions about the distributions of the input variables and can use both numeric

and categorical data (Quinlan, 1996, Mitchell, 1997, Rokach and Maimon, 2005, Tan et al., 2014). Additionally, they effectively handle nonlinear relationships and complex interactions (Strobl et al., 2009).

However, a disadvantage of decision tree models is that they do not explicitly take into account spatial trends in the data. Unlike geostatistical methods, such as kriging, the predictions can therefore contain spatial bias.

A number of studies have applied regression-kriging (RK) as a solution (Knotters et al., 1995, Odeh et al., 1995, Hengl et al.,

2004). By kriging the residuals of the predictive model and adding the kriged residuals to the prediction surface, this approach can account for spatial trends and achieve higher accuracies. A disadvantage of RK is that the combination of two models hinders the combination of spatial trends with the other covariates. Spatial trends therefore remain disconnected from other statistical relationships in the analysis, leading to difficulties in interpreting the model and its associated uncertainties.

An obvious solution to this problem would be to use the x- and y-coordinates of the soil observations as covariates.

However, results have shown that this approach can lead to unrealistic orthogonal artefacts in the output maps when used in conjunction with decision tree algorithms (Behrens et al., 2018, Hengl et al., 2018, Nussbaum et al., 2018). The cause of this problem lies in the splitting procedure of decision tree algorithms, as they use only one covariate for each split. Therefore, a dataset containing only the x- and y-coordinates will force the algorithm to make orthogonal splits in geographic space. Several researchers have proposed solutions to this problem. Behrens et al. (2018) proposed the use of Euclidean distance

fields (EDF) in the form of distances to the corners and middle of the study area as well as the x- and y-coordinates. Their results showed that this approach efficiently integrated spatial trends and that accuracies were better than or on par with other methods for integrating spatial context.

On the other hand, Hengl et al. (2018) suggested an approach referred to as spatial Random Forest (RFsp). This method consists of calculating data layers with buffer distances to each of the soil observations in the training dataset. It then trains a

Random Forest model, using the buffer distances as covariates, combined with auxiliary data or on their own. One of the main advantages of this approach is that it incorporates distances between observations in a similar manner to geostatistical models. The authors assessed the use of RFsp on a large number of spatial prediction problems and showed that it effectively eliminated spatial trends in the residuals.

Although these two methods are able to integrate spatial trends in machine learning models, they can be difficult to interpret.

The distances used in EDF depend on the geometry of the study area, and for RFsp, they depend on the locations of the soil samples. The meaning and interpretation of the distances therefore varies depending on the study area and the soil observations.

EDF and RFsp also have limited flexibility, as both methods specify the number of geographic data layers a priori. For EDF, the number of distance fields is seven, and for RFsp, the number of buffer distances is equal to the number of soil

observations. This means that there is no straightforward way to increase the number of spatially explicit covariates, if the number is insufficient to account for spatial trends. Vice versa, there is no way to decrease the number of spatially explicit covariates, even if a smaller number would suffice. The latter is especially relevant for RFsp, as the method is computationally unfeasible for datasets with a large number of observations (Hengl et al., 2018).

In this study, we propose an alternative method for including spatially explicit covariates for mapping soil properties. With

the method, we aim to address directly the cause of the orthogonal artefacts produced with x- and y-coordinates as covariates in decision tree models. Furthermore, we aim to improve upon the shortcomings of previous methods by developing a method that is both flexible and easily interpretable.

We refer to the method as Oblique Geographic Coordinates (OGC). In short, it works by calculating coordinates for the observations along a series of axes, tilted at several oblique angles relative to the x-axis. By including oblique coordinates as

covariates, we enable the decision tree algorithm to make oblique splits in geographic space. As this is not possible with only x- and y-coordinates as covariates, this addition should allow the model to produce a more realistic prediction surface. Furthermore, the number of oblique angles is adjustable, and soil mappers can therefore choose a number that suits their

purpose. Some mapping tasks may require a higher number of oblique angles than others, and soil mappers can therefore increase the number as necessary. Alternatively, if a small number of oblique angles suffices, soil mappers can reduce their number and thereby shorten computation times.

We test the method on four spatial datasets. Firstly, we test it for predicting soil organic matter contents in a densely sampled agricultural field in Denmark, located in northern Europe. Secondly, we test it on three publicly available spatial datasets (*meuse*, *eberg* and Swiss rainfall). We hypothesise that OGC can provide accuracies on par with previous methods for including explicitly spatial covariates. We also hypothesize that it is possible to adjust the number of oblique angles in order to optimize accuracy, and that the results allow meaningful interpretations.

## 2 Materials and methods

### 2.1 Study areas

We test OGC and compare it to other methods based on four spatial datasets. Firstly, we test it for a predicting soil organic matter (SOM) for an agricultural field in Denmark (Vindum). Secondly, we test it on three publicly available datasets. For Vindum, we will present methods and results in detail. For the other three datasets, we will present methods and results in brief, while Appendix A contains a detailed presentation of the methods and results for these datasets.

#### 2.1.1 Vindum

This study area is a 12-ha agricultural field located in Denmark in northern Europe (9.568°E; 56.375°N, ETRS 1989) (Figure 1). It lies in a kettled moraine landscape 55 – 66 m above sea level. The parent materials in the field include clay till, glaciofluvial sand and peat. The climate is temperate coastal, with mean monthly temperatures ranging from 1°C in January to 17°C in July and a mean annual precipitation of 850 mm (Wang, 2013). The field contains 285 measurements of soil organic matter (SOM) from the depth interval 0 – 25 cm, located in a 20 m grid.

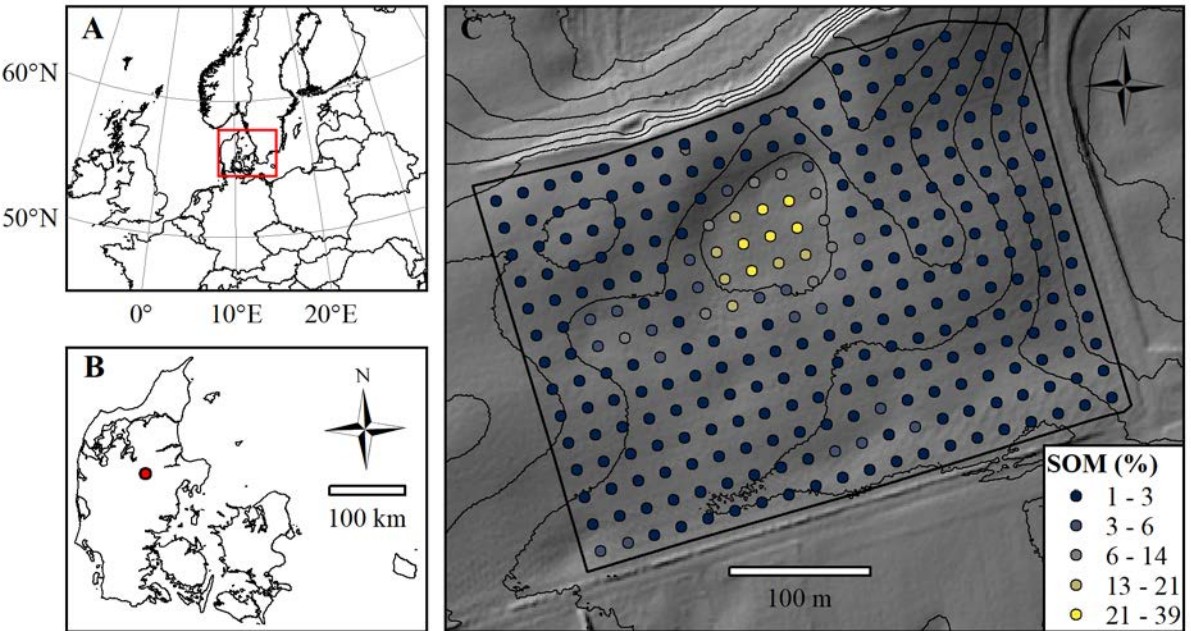

**Figure 1: A: Location of Denmark in northern Europe. B: Location of the Vindum field within Denmark. C: Map of the Vindum field, including locations of the samples extracted for soil organic matter (SOM) measurements. The thin black lines are 2 m contour lines. The background shows hill shade (northwest, 45° altitude) based a digital elevation model (DEM) in 1.6x1.6 m resolution (National Survey and Cadastre, 2012).**

The SOM contents of the topsoil in the field range from 1.3% to 38.8% with a mean value of 3.5% and a median of 2.2%.

The values have a strong positive skew of 4.7 and are leptokurtic with a kurtosis of 26.9. Logarithmic transformation reduces skewness (2.9) and kurtosis (11.1). Pouladi et al. (2019) described that spatial structure of the data with a stable variogram with 139 m range, nugget of 0 and sill of 23.8.

### 2.1.2 Additional datasets

For additional analyses, we included the *meuse* dataset, the *eberg* dataset and the Swiss rainfall dataset. The *meuse* dataset,

available through the R package *sp* (Pebesma et al., 2020), contains 155 measurements of soil heavy metal concentrations from a 5-km$^2$ flood plain of the Meuse river near the village of Stein in the Netherlands. For this dataset, we mapped zinc concentrations. The *eberg* dataset, available through the R package *plotKML* (Hengl et al., 2020) contains 3,670 soil observations from a 100-km$^2$ area in Ebergötzen near the city Göttingen in Germany. For this dataset, we mapped soil types. Lastly, the *Swiss rainfall* dataset contains 476 rainfall measurements from May 8, 1986 in Switzerland (Dubois et al., 2003).

Although this is not a soil dataset, we included it because of the high anisotropy of the data, which makes it useful for comparing methods on their ability to account for anisotropic spatial problems. We describe these three datasets in more detail in Appendix A.

## 2.2 Oblique geographic coordinates

The method that we propose consists of calculating coordinates along a number of axes titled at various oblique angles, relative to the x-axis. In the following, we show that it is possible to calculate the coordinate of a point $(b_1, a_1)$ along an axis tilted at an angle $\theta$ relative to the x-axis, based on $\theta$ and the x- and y-coordinates of $(b_1, a_1)$. Equation 1 shows the the calculation of the oblique geographic coordinate, using Figure 2 for illustration.

$$\text{OGC} = b_2 = \sqrt{a_1{}^2 + b_1{}^2} * \cos\left(\theta - \tan^{-1}\frac{a_1}{b_1}\right) \tag{1}$$

, where $\theta$ is the angle of the titled axis relative to the x-axis; $a_1$ is the y-coordinate of $(b_1, a_1)$; $b_1$ is the x-coordinate of $(b_1, a_1)$; $b_2$ (or "$OGC$") is the coordinate of $(b_1, a_1)$ along an axis tilted with the angle $\theta$ relative to the x-axis.

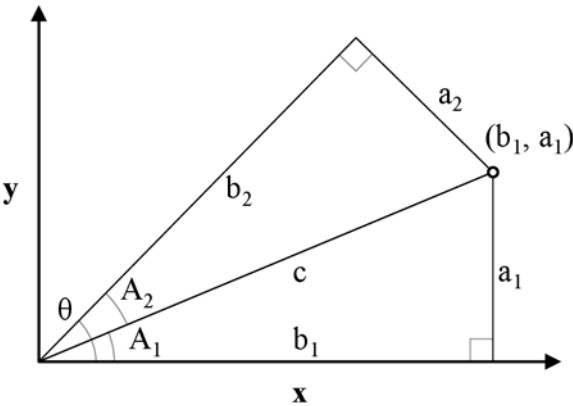

**Figure 2: Illustration for the derivation of the oblique geographic coordinate for the point $(b_1, a_1)$ along an axis tilted with the angle $\theta$ from the x-axis. The coordinate is equal to the length of $b_2$. Triangles $a_1b_1c$ and $a_2b_2c$ are right triangles with the same hypotenuse $c$. The sides $a_1$ and $b_1$ are the x- and y-coordinates of the point $(b_1, a_1)$, respectively. $A_1$ is the angle between the x-axis and the line $c$ between the origin of the coordinate system and the point $(b_1, a_1)$; $A_2$ is the difference between $\theta$ and $A_1$.**

As the x- and y-coordinates of soil observations are known, and $\theta$ is given, it is possible to calculate coordinates at oblique angles for all soil observations in a dataset. Likewise, as the x- and y-coordinates of the cells in a geographic raster layer are known, it is possible to calculate oblique coordinates for the cells. Our approach relies on calculating coordinates along $n$ axes tilted at angles ranging from 0 to $\pi((n-1)/n)$ with increments of $\pi/n$ between the angles. $\theta$ should not be $\pi$ or greater, as coordinates along axes tilted at these angles will correlate with coordinates along axes tilted at angles of 0 to $\pi((n-1)/n)$. For example, coordinates along an axis with $\theta = 0.25\pi$ (northeast) perfectly correlate with coordinates along an axis with $\theta = 1.25\pi$ (southwest). Figure 3 shows coordinates along axes tilted at six different angles relative to the x-axis for the Vindum study area. The coordinate rasters A and D are equivalent to the x- and y-coordinates, respectively, while the coordinate rasters B, C, E and F show coordinates at oblique angles.

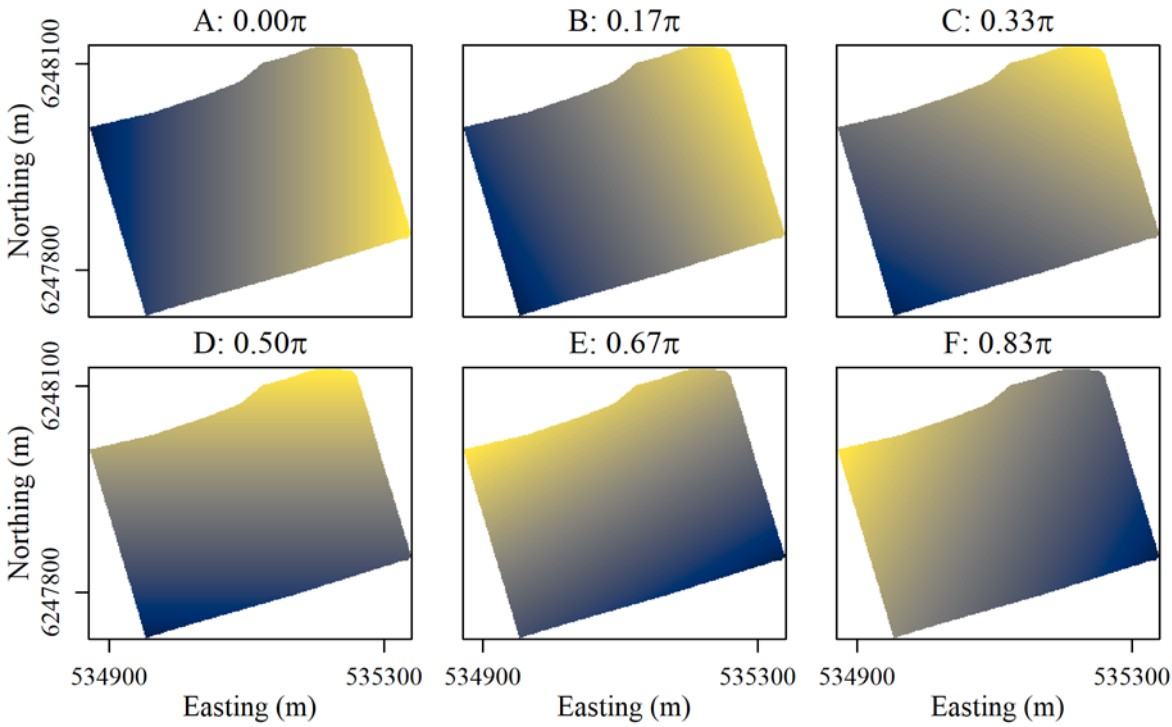


**Figure 3: Examples of rasters with coordinates tilted at six different angles for the Vindum study area. Easting and northing for UTM Zone 32N, ETRS 1989.**

## 2.3 Method comparison

### 2.3.1 Vindum

We use the 285 SOM observations from the Vindum study area in order to test the accuracy of predictions made by Random Forest models using OGC as covariates. In addition to OGC, we also employed 19 data layers with auxiliary data, which Pouladi et al. (2019) derived from a 1.6 m DEM, satellite imagery and electromagnetic induction. Topographic variables included the sine and cosine of the aspect, depth of sinks, plan and profile curvature, elevation, flow accumulation, valley bottom flatness, mid-slope position, standard and modified topographic wetness index, slope gradient, slope length and

valley depth. Satellite imagery included normalized difference, absolute difference, ratio and soil-adjusted vegetation indices. Lastly, we used the apparent electrical conductivity from a DUALEM 1 sensor in perpendicular mode.

**Table 1: Auxiliary data variables used as covariates in the study, including name, description, the mean value and the range. Pouladi et al. (2019) describe the derivation of the variables.**

| Predictor variable | Description | Mean (range) |
|---|---|---|
| **DEM** | | |
| Cos aspect | Cosine of surface aspect | -0.1 (-1.0 - 1.0) |

| | | |
|---|---|---|
| Sin aspect | Sine of surface aspect | 0.32 (-1.0 - 1.0) |
| Depth of sinks | Depth of sinks (m) | 0.1 (0.0 - 1.1) |
| Plan curvature | Shape of the surface in the horizontal plane | 0 (-34 - 15) |
| Profile curvature | Shape of the surface in the vertical plane | 0.00 (-0.06 - 0.04) |
| Elevation | Elevation from DEM; m above sea level | 60.8 (54.6 - 66.2) |
| Flow accumulation | Number of upslope cells | 74 (3 - 8969) |
| MRVBF | Multiresolution index of valley bottom flatness | 1.5 (0.0 - 4.9) |
| Mid-slope position | Covers the warmer zones of slopes | 0.5 (0.0 - 1.0) |
| SAGA wetness index | SAGA GIS modified topographic wetness index | 4.0 (2.2 - 8.6) |
| Slope gradient | Local slope gradient (degrees) | 4.9 (0.0 - 17.5) |
| SL | Slope length factor | 0.4 (0.0 - 2.3) |
| TWI | Topographic wetness index | 6.6 (3.7 - 14.6) |
| Valley depth | Depth of valleys (m) | 1.4 (0.1 - 8.1) |
| **Sentinel 2** | | |
| DVI | Difference vegetation index | 1735 (1202 - 3294) |
| NDVI | Normalized difference vegetation index | 0.5 (0.3 - 0.7) |
| RVI | Ratio vegetation index | 2.8 (2.0 - 6.4) |
| SAVI | Soil-adjusted vegetation index | 0.7 (0.5 - 1.1) |
| **DUALEM 1mPRP** | | |
| ECa | Apparent electrical conductivity | 8.9 (4.9 - 16.0) |


In order to optimize the number of raster layers for OGC, we generated datasets with 2 – 100 coordinate rasters. We then trained Random Forest models from each dataset, both with and without auxiliary data. In order to assess predictive accuracy, we used 100 repeated splits on the SOM observations, each using 75% of the observations for model training and a 25% holdout dataset for accuracy assessment. We trained models using the R package *ranger* (Wright and Ziegler, 2015)

and parameterized the models using the R package *caret* (Kuhn, 2008). For each split, we tested five different values for *mtry*, minimum node sizes of 1, 2, 4 and 8, and two different splitting rules *variance* and *extratrees*. We mainly adjusted *mtry* and the minimum node size in order to avoid overfitting. We tested *mtry* values at even intervals between 2 and the total number of covariates, including both auxiliary data and spatially explicit covariates. The tested *mtry* values therefore varied depending on the number of covariates. The *extratrees* splitting rule generates random splits, as opposed to the

*variance* splitting rule, which chooses optimal splits. Per default, *extratrees* generates one random split for each covariate and then chooses the random split that gives the largest variance reduction (Geurts et al., 2006). It therefore leads to a greater degree of randomization. We selected the setup that provided the lowest RMSE for the out-of-bag predictions on the training data, and used this setup for predictions on the 25% holdout dataset.

We used the same 100 repeated splits for each number of coordinate rasters, with and without auxiliary data. We calculated accuracy based on Pearson's $R^2$, RMSE and Lin's concordance criterion (ccc), and subsequently used the number of coordinate rasters that yielded the lowest RMSE. We selected a different number of coordinate rasters with and without auxiliary data.

We then compared the accuracies obtained with the optimal numbers of coordinate rasters, with and without auxiliary data, to the accuracies obtained with other methods. We tested kriging, Random Forest models trained only on the auxiliary data and Random Forest models trained using EDF and RFsp, with and without auxiliary data. We trained the Random Forest models using the same procedure outlined above. For kriging, we used variograms automatically fitted on logarithmic-transformed SOM observations using the *autofitVariogram* function of the R package *automap* (Hiemstra, 2013). A previous study using the same dataset showed that kriging predicted SOM more accurately than regression-kriging (Pouladi et al., 2019). We therefore omitted regression-kriging from the analysis, although, without this previous finding, it would have been relevant to include it.

We used the same 100 repeated splits for assessing the accuracies of all methods. This allowed us to carry out pairwise t-tests between the accuracies of the methods. We used the results of the pairwise t-tests to rank the methods according to their accuracies according to each of the metrics. If there was no statistical difference ($p > 0.05$) between the accuracies of two or more methods, these methods received the same rank. We calculated separate ranks for the methods for each accuracy metric, resulting in three different sets of ranks. In order to illustrate the results, we produced maps of SOM with each method, using models trained from all the data.

We also investigated the covariate importance of models trained with OGC and tested all methods for spatially autocorrelated residuals using experimental variograms. To produce sample variograms of the residuals, we produced maps with each method using all observations. We converted both observations and predictions to natural logarithmic scale. We then subtracted the predictions from the observations and calculated variograms for these residuals. For this purpose, we used the function *variogram* from the R package *gstat* (Pebesma and Graeler, 2020) with its default parameters.

### 2.3.1 Additional datasets

We also compared OGC to other methods based on the three additional datasets *meuse*, *eberg* and Swiss rainfall. The methods in the comparison depended on the dataset. For the *meuse* dataset, we tested all the methods tested on the Vindum dataset, with the addition of RK using Random Forest models for regression. For the *eberg* dataset, we tested Random Forest models based on auxiliary data (AUX), EDF and OGC, as well as the combined methods (EDF + AUX and OGC + AUX). For the Swiss rainfall dataset, we tested only purely spatial methods, including ordinary kriging (OK), EDF, RFsp and OGC. As for the Vindum dataset, we tested each method based on 100 splits into training and test data and carried out pairwise t-tests on the resulting accuracies. Appendix A gives additional details on the methods for each dataset. For the three additional datasets, we focused on the accuracies and maps produced with each method. We therefore omitted analyses of the residuals and covariate importance for these datasets.

# 3 Results and discussion

## 3.1 Optimal number of coordinate rasters

### 3.1.1 Vindum

For the Vindum dataset, accuracies of predictions obtained with OGC, without auxiliary data, increased with the number of coordinate rasters up to an optimum at seven coordinate rasters (Figure 4). However, with more than seven coordinate rasters, accuracies deteriorated slightly with the number of coordinate rasters. This pattern was the same for all three metrics. On the other hand, with OGC in combination with auxiliary data, accuracies generally increased with the number of coordinate rasters. The increase was greatest when the number of coordinate rasters was small, while the effect of more

coordinate rasters decreased for larger numbers of coordinate rasters. With auxiliary data, the optimal number of coordinate rasters was 94 for Pearson's $R^2$, 80 for RMSE and 89 for ccc. Accuracies with auxiliary data were almost invariably higher than accuracies achieved without auxiliary data.

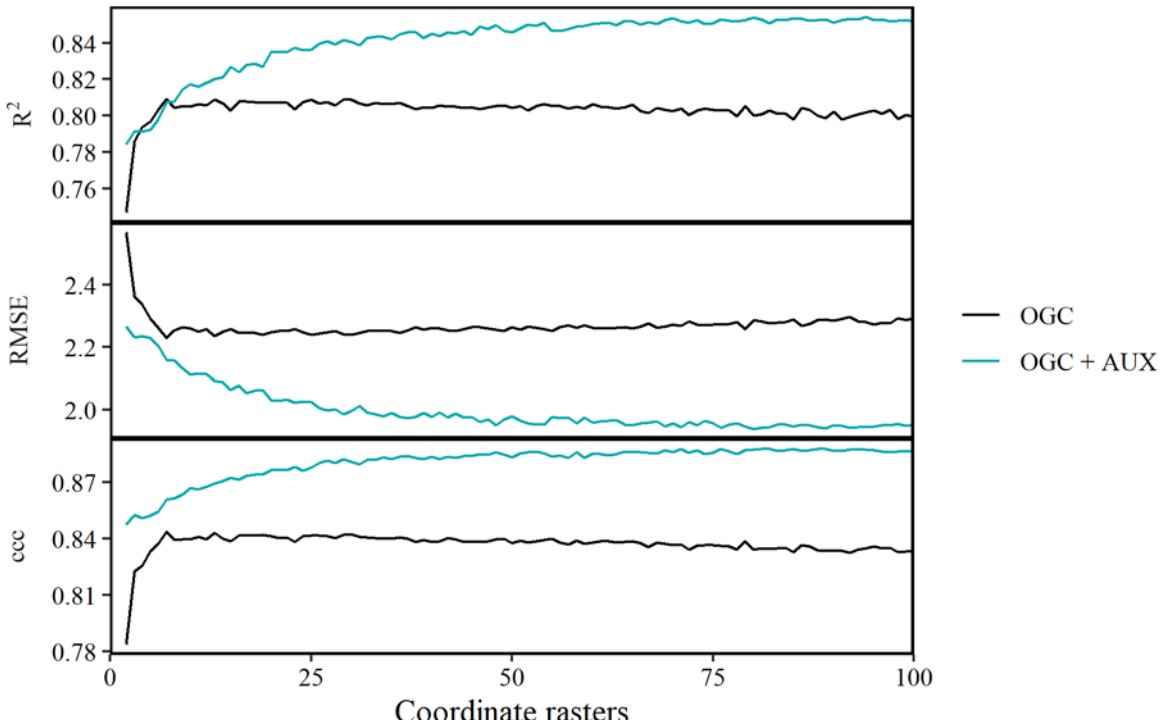

**Figure 4: Effects of the number of coordinate rasters on the accuracy of SOM predictions on the Vindum dataset, calculated as**
**Pearson's $R^2$, root mean square error (RMSE) and Lin's concordance criterion (ccc). We calculated effects for Random Forest models trained on only coordinate rasters (OGC) and with coordinate rasters in combination with auxiliary data (OGC + AUX). The lines represent mean values obtained from 100 repeated splits (75% training dataset, 25% test dataset) for each number of coordinate rasters.**

Figure 5 shows SOM contents mapped for Vindum with increasing numbers of coordinate rasters, without auxiliary data.
The predictions with only two coordinate rasters showed a pattern very typical of predictions with x- and y-coordinates with very visible orthogonal artefacts. As the number of coordinate rasters increased, the patterns of the artefacts changed. With coordinate rasters at three different angles, the artefacts had a hexagonal pattern, and with coordinate rasters at four different angles, the artefacts gained an octagonal pattern. Furthermore, as the number of coordinate rasters increased, the artefacts became less pronounced. Although some artefacts were visible with coordinate rasters at seven different angles, they were much less visible than the artefacts in the map produced with only two coordinate rasters.

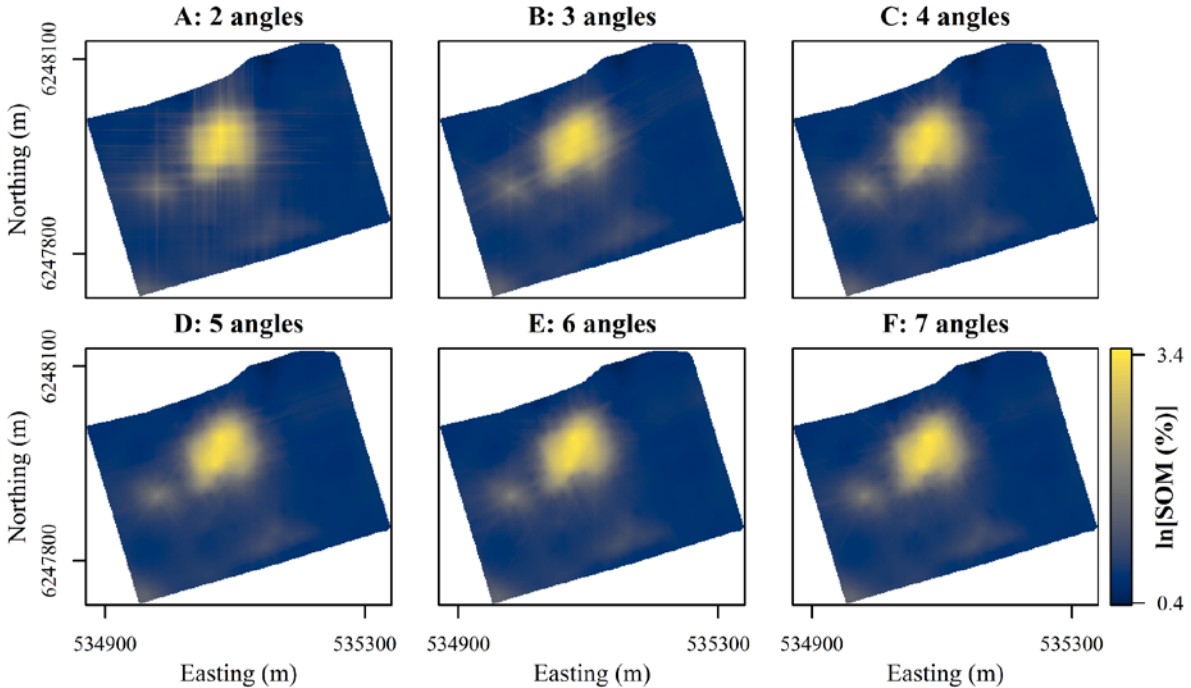

**Figure 5: Maps of soil organic matter (SOM) contents in the topsoil at Vindum predicted using Random Forest models trained with coordinate rasters at two to seven different angles as covariates. Easting and northing for UTM Zone 32N, ETRS 1989.**

With auxiliary data, the effect of increasing the number of coordinate rasters was less clearly visible for the Vindum dataset (Figure 6). Even with only two coordinate rasters, the predictions had no orthogonal artefacts. However, they contained noisy patterns and sharp boundaries in some areas. This is most likely an artefact from the auxiliary data. For example, using a high-resolution DEM may have created noise in the predictions. However, with coordinate rasters at 80 different angles, the spatial pattern of the predicted SOM contents became substantially smoother, with a reduction both in noise and in sharp boundaries. Furthermore, some areas with moderately high SOM contents became more clearly visible and coherent, for example in the area approximately one third of the way from the western to the northern corner of study area. The predicted patterns with a higher number of coordinate rasters were therefore not only more accurate, but also more realistic.

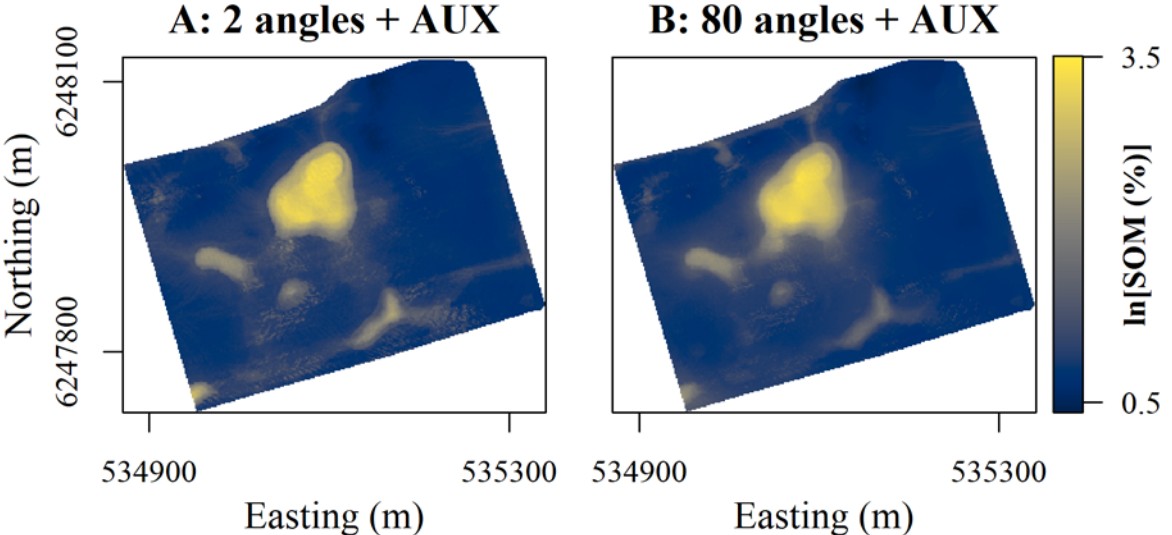

Figure 6: Maps of soil organic matter (SOM) contents in the topsoil at Vindum predicted using Random Forest models trained using auxiliary data in conjunction with coordinate rasters at (A) two and (B) 80 different angles as covariates. Easting and northing for UTM Zone 32N, ETRS 1989.

### 3.1.2 Additional datasets

For the three additional datasets, the effect of increasing the number of coordinate rasters without auxiliary data was generally the same as for the Vindum dataset. In all three cases, there was relatively little, if any increase in accuracy after an initially very steep increase. For the *meuse* dataset, the optimal number of coordinate rasters was six or eight, depending on the accuracy metric (Figure A1). For the *eberg* dataset, the optimal number was 91, but there was only limited improvement in accuracy with more than five coordinate rasters (Figure A3). For the Swiss rainfall dataset, the optimal number of coordinate rasters was 33 or 50, depending on the accuracy metric (Figure A5).

As for the Vindum dataset, the optimal number of coordinate rasters was generally larger in combination with auxiliary data than without auxiliary data. For the *meuse* dataset, the optimal number of coordinate rasters in combination with auxiliary data was 11 or 13, depending on the accuracy metric. For the *eberg* dataset, the optimal number of coordinate rasters in combination with auxiliary data was 22. However, unlike the results for the Vindum dataset, accuracies for these two datasets gradually decreased when the number of coordinate rasters was larger than the optimal value.

In summary, the combination of OGC with auxiliary data generally increased the optimal number of coordinate rasters. Furthermore, in some cases, accuracy deteriorates when the number of coordinate rasters surpasses an optimal value, while in other cases it reaches a plateau. The decrease in accuracy past the optimum may be due to correlation between the coordinate rasters. Coordinates x and y are perfectly uncorrelated, but the coordinate rasters become increasingly correlated as their number increases. The optimal value may therefore be a trade-off between the increased ability of the model to

account for spatial trends and the adverse effect of increasingly correlated covariates. It is therefore likely that it depends on
the complexity of the spatial distribution of the target variable as well as the number of observations.

With OGC in combination with auxiliary data, the process-based covariates in the auxiliary data most likely help to reduce the effect of correlation between the coordinate rasters. Furthermore, in this case, the number of coordinate rasters also affects the relative weighting between the auxiliary data and the coordinate rasters. When *mtry* is smaller than the total number of covariates, a higher number of coordinate rasters increases the chance that a coordinate raster will be available for
a split. The optimal number of coordinate rasters may therefore depend on the optimal weighting between process-based and explicitly spatial covariates. This optimal weighting may depend on the number of covariates in the auxiliary data as well as the strength of the relationship between the target variable and the auxiliary data.

At present, several factors could therefore explain the optimal number of coordinate rasters for each dataset, with and without auxiliary data. The exact interplay between these factors is unclear, and the best option may therefore be to
experiment with different numbers of coordinate rasters.

## 3.2 Method comparison

### 3.2.1 Predictive accuracy

For all four datasets, there were large overlaps in the accuracies of the methods, as accuracies varied across the 100 repeated splits (Figure 7, Figure A2, Figure A4, Figure A6). However, an analysis on the Vindum dataset revealed that the accuracies
generally correlated between the methods across the repeated splits. The mean correlation coefficient (Pearson's R) was 0.52 (0.19 – 0.88) for $R^2$, 0.71 (0.65 – 0.71) for RMSE and 0.65 (0.41 – 0.89) for ccc. This shows that some holdout datasets yielded consistently high accuracies, while others yielded consistently low accuracies. Furthermore, especially for $R^2$ and ccc, a few holdout datasets yielded much lower accuracies than the other holdout datasets, leading to long negative tails (Figure 7, Figure A2, Figure A6).


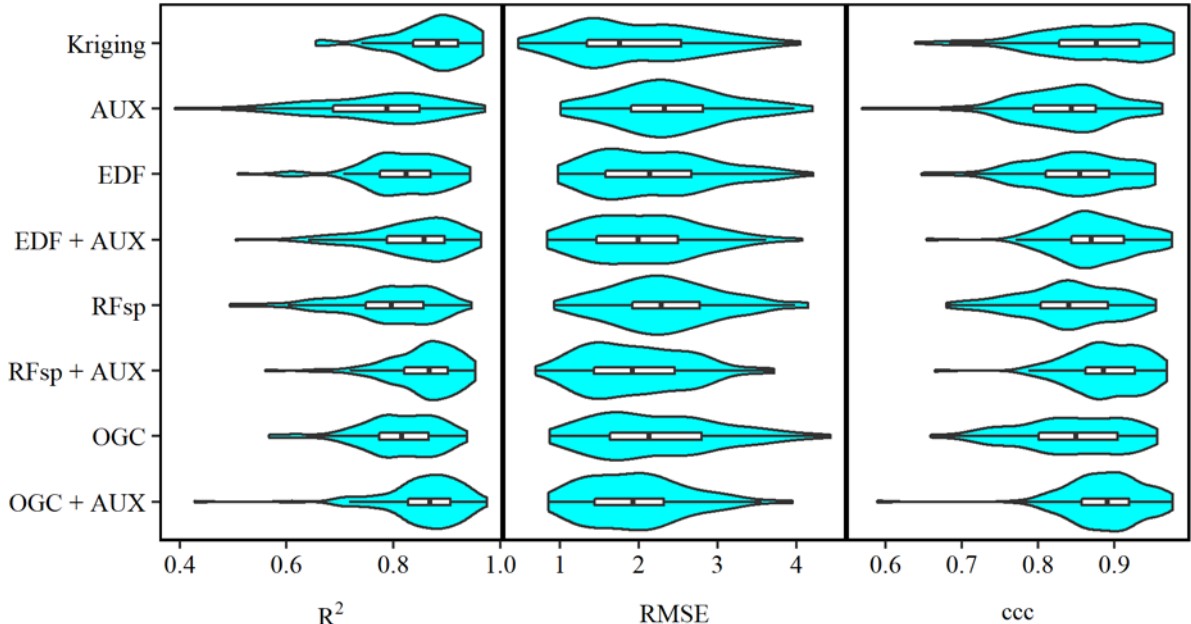

**Figure 7: Violin plots showing accuracies of soil organic matter predictions on the Vindum dataset with kriging, and Random Forest models trained using either auxiliary data (AUX), Euclidean distance fields (EDF), distances to observations (RFsp), oblique geographic coordinates (OGC) or EDF, RFsp or OGC in conjunction with AUX. The figure shows Pearson's $R^2$, room mean square error (RMSE) and Lin's concordance obtained from 100 repeated splits (75% training dataset, 25% test dataset).**

For the Vindum dataset, kriging achieved the highest rank for $R^2$ (Table 2). For RMSE, kriging shared the highest rank with EDF, RFsp and OGC in combination with auxiliary data. Lastly, OGC and RFsp in combination with auxiliary data shared the highest rank for ccc. In short, kriging, RFsp with auxiliary data and OGC with auxiliary data all had the highest rank for two accuracy metrics out of three. We therefore regard these three methods as the most accurate methods for the Vindum dataset. Furthermore, we regard these three methods as equally accurate for this dataset, as none of them was universally more accurate than the other two methods.

Auxiliary data used on their own, as well as RFsp without auxiliary data had the lowest rank for all three accuracy metrics on the Vindum dataset. Furthermore, OGC without auxiliary data had the same rank as EDF without auxiliary data for all three accuracy metrics.

**Table 2: Ranks for the accuracies of the methods on the Vindum dataset, calculated as Pearson's $R^2$, RMSE and ccc, respectively. Methods for which a pairwise t-test did not give a significant difference in accuracy ($p > 0.05$) received equal ranks for the metric in question. Ranks for the methods therefore differ between the three metrics. AUX: Auxiliary data. EDF: Euclidean distance fields. OGC: Oblique geographic coordinates. RFsp: Distances between observations.**

| Rank | $R^2$ | | RMSE | | ccc | |
|---|---|---|---|---|---|---|
| | Method | Mean | Method | Mean | Method | Mean |
| 1 | Kriging | 0.87 | EDF + AUX | 2.0 | OGC + AUX | 0.89 |
| | | | Kriging | 2.0 | RFsp + AUX | 0.89 |
| | | | OGC + AUX | 1.9 | | |
| | | | RFsp + AUX | 1.9 | | |
| 2 | OGC + AUX | 0.85 | EDF | 2.2 | EDF + AUX | 0.87 |
| | RFsp + AUX | 0.86 | OGC | 2.2 | Kriging | 0.87 |
| 3 | EDF | 0.82 | AUX | 2.4 | AUX | 0.84 |
| | EDF + AUX | 0.83 | RFsp | 2.3 | EDF | 0.85 |
| | OGC | 0.81 | | | OGC | 0.84 |
| | | | | | RFsp | 0.84 |
| 4 | AUX | 0.77 | | | | |
| | RFsp | 0.79 | | | | |


Pouladi et al. (2019) tested several methods for predicting SOM on the Vindum dataset, including kriging and the machine learning algorithms Cubist and Random Forest, with and without kriged residuals. The authors found that kriging provided the most accurate predictions of SOM. The results for Vindum affirm the high accuracy of kriging predictions, but they also show that Random Forest models combining auxiliary data with spatial trends can achieve similar accuracies.

For the *meuse* dataset, OGC in combination with auxiliary data achieved the highest rank for $R^2$ and RMSE (Table 3). For ccc, OGC in combination with auxiliary data shared the highest rank with EDF in combination with auxiliary data. Without auxiliary data, OGC received third rank for RMSE and fourth rank with $R^2$ and ccc. OGC without auxiliary data was generally on par with models based only on auxiliary data and with EDF. It was less accurate than combined methods and OK ($R^2$ and ccc). RFsp without auxiliary data was the least accurate method.

**Table 3: Ranked accuracies obtained with each method on the *meuse* dataset, calculated as Pearson's $R^2$, RMSE and ccc. Methods received shared ranks if a pairwise t-test revealed no statistically significant difference between their accuracies for the metric in question. Each t-test used accuracies obtained with 100 repeated splits into training and test datasets. AUX: Auxiliary data. EDF: Euclidean distance fields. RFsp: Distances to observations. OGC: Oblique geographic coordinates. OK: Ordinary kriging. RK: Regression-kriging.**

| Rank | $R^2$ | | RMSE | | ccc | |
|---|---|---|---|---|---|---|
| | Method | Mean | Method | Mean | Method | Mean |
| 1 | OGC + AUX | 0.68 | OGC + AUX | 202 | EDF + AUX | 0.78 |
| | | | | | OGC + AUX | 0.78 |
| 2 | EDF + AUX | 0.67 | EDF + AUX | 204 | RFsp + AUX | 0.77 |

| Rank | Method | | Method | | Method | |
|---|---|---|---|---|---|---|
| | RFsp + AUX | 0.66 | RFsp + AUX | 206 | | |
| 3 | OK | 0.63 | AUX | 224 | OK | 0.76 |
| | RK | 0.63 | EDF | 226 | RK | 0.76 |
| | | | OGC | 220 | | |
| | | | OK | 215 | | |
| | | | RK | 216 | | |
| 4 | AUX | 0.61 | RFsp | 250 | AUX | 0.74 |
| | EDF | 0.59 | | | OGC | 0.74 |
| | OGC | 0.61 | | | | |
| 5 | RFsp | 0.50 | | | EDF | 0.71 |
| 6 | | | | | RFsp | 0.63 |


For the *eberg* dataset, OGC in combination with auxiliary data was the most accurate method (Table 4). Without auxiliary data, OGC had the third rank. It was less accurate than EDF combined with auxiliary data, but more accurate than EDF without auxiliary data and models based only on auxiliary data. Models based only on auxiliary data yielded the lowest accuracies.

**Table 4: Ranks of the accuracies (percent cases correctly predicted) obtained with each method on the *eberg* dataset. Pairwise t-tests showed that differences between the accuracies of the methods were all statistically significant (p < 0.05). AUX: Auxiliary data. EDF: Euclidean distance fields. OGC: Oblique geographic coordinates.**

| Rank | Method | Accuracy |
|---|---|---|
| 1 | OGC + AUX | 0.39 |
| 2 | EDF + AUX | 0.38 |
| 3 | OGC | 0.37 |
| 4 | EDF | 0.37 |
| 5 | AUX | 0.35 |

For the Swiss rainfall dataset, OGC was the most accurate method for all three metrics (Table 5). RFsp was the second-most

accurate method, followed by EDF. OK was the least accurate method.

**Table 5: Ranked accuracies on the Swiss rainfall dataset for each method. Pairwise t-test showed statistically significant (p < 0.05) differences between the methods for all three metrics. The ranks are the same for all three metrics. EDF: Euclidean distance fields. RFsp: Distances to observations. OGC: Oblique geographic coordinates. OK: Ordinary kriging.**

| Rank | Method | $R^2$ | RMSE | ccc |
|---|---|---|---|---|
| 1 | OGC | 0.831 | 4.7 | 0.902 |
| 2 | RFsp | 0.822 | 4.8 | 0.893 |
| 3 | EDF | 0.818 | 4.9 | 0.891 |
| 4 | OK | 0.804 | 5.0 | 0.887 |

In summary, for Vindum, *meuse* and *eberg*, OGC combined with auxiliary data was either the most accurate method or one of the most accurate methods. Without auxiliary data, OGC was not one of the most accurate methods for these datasets. However, for the Swiss rainfall dataset, OGC was the most accurate method, even though we used no auxiliary data.

It is important to consider that in most cases all methods yielded acceptable accuracies. Although the differences between the accuracies of the methods were in many cases statistically significant, they were generally small. However, the results show

that OGC compares well with other methods for integrating spatial trends in machine learning models.

### 3.2.2 Maps

For the Vindum dataset, kriging produced a smooth prediction surface, which is very common for this method (Figure 8A). The prediction surface with EDF was mostly smooth, but it also contained a distinct "rings in the water" artefact caused by the raster with the distance to the middle of the study area (Figure 8B). The prediction surface with RFsp was smoother than

the prediction surface produced by kriging (Figure 8C). The predictions with only auxiliary data were very similar to the predictions made with x- and y-coordinates in combination with auxiliary data (compare Figure 8C to Figure **6**A). In combination with auxiliary data, both EDF and RFsp produced smoothing effects similar to the effect seen with OGC in combination with auxiliary data (compare Figure 8E and Figure 8F to Figure **6**B). However, for EDF the smoothing was less visible than with OGC, and for RFsp it was more visible than with OGC.

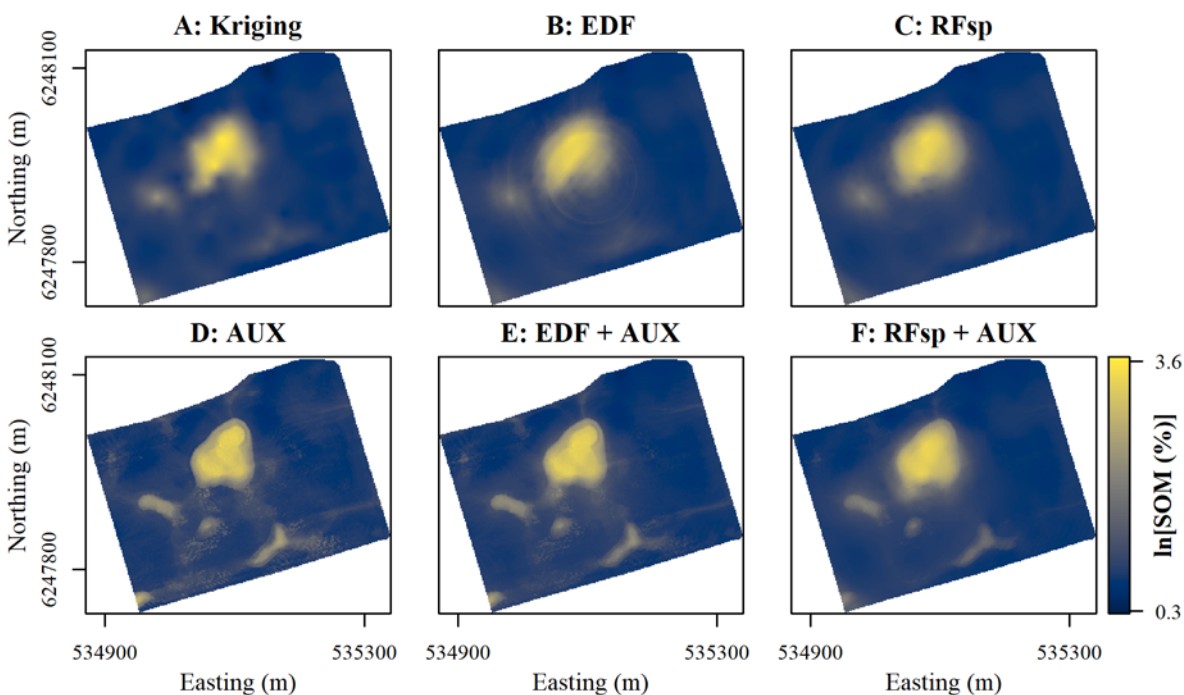


**Figure 8: Prediction of soil organic matter (SOM) contents for the topsoil at Vindum using A: Kriging, or Random Forest models trained with B: Euclidean distance fields (EDF), C: Distances to observations (RFsp), D: Auxiliary data (AUX), E: EDF in conjunction with AUX, or F: RFsp in conjunction with AUX. Easting and northing for UTM Zone 32N, ETRS 1989.**

For the *meuse* dataset, OK, EDF and RFsp produced smooth prediction surfaces (Figure 9). However, OGC without auxiliary data produced a prediction surface with several abrupt, angular artefacts. The accuracy of OGC without auxiliary data was on par with some of the other methods, but the maps revealed that the predictions were not realistic. Predictions with the combined methods (RK, EDF + AUX, RFsp + AUX and OGC + AUX) were mostly similar to predictions with only auxiliary data. However, in some places these methods smoothed out the spatial patterns produced with only auxiliary data (for example in the northern part of the study area), and in other places they made them more distinct (for example south-west of the middle of the study area). In this regard, the results are similar to the results from Vindum.

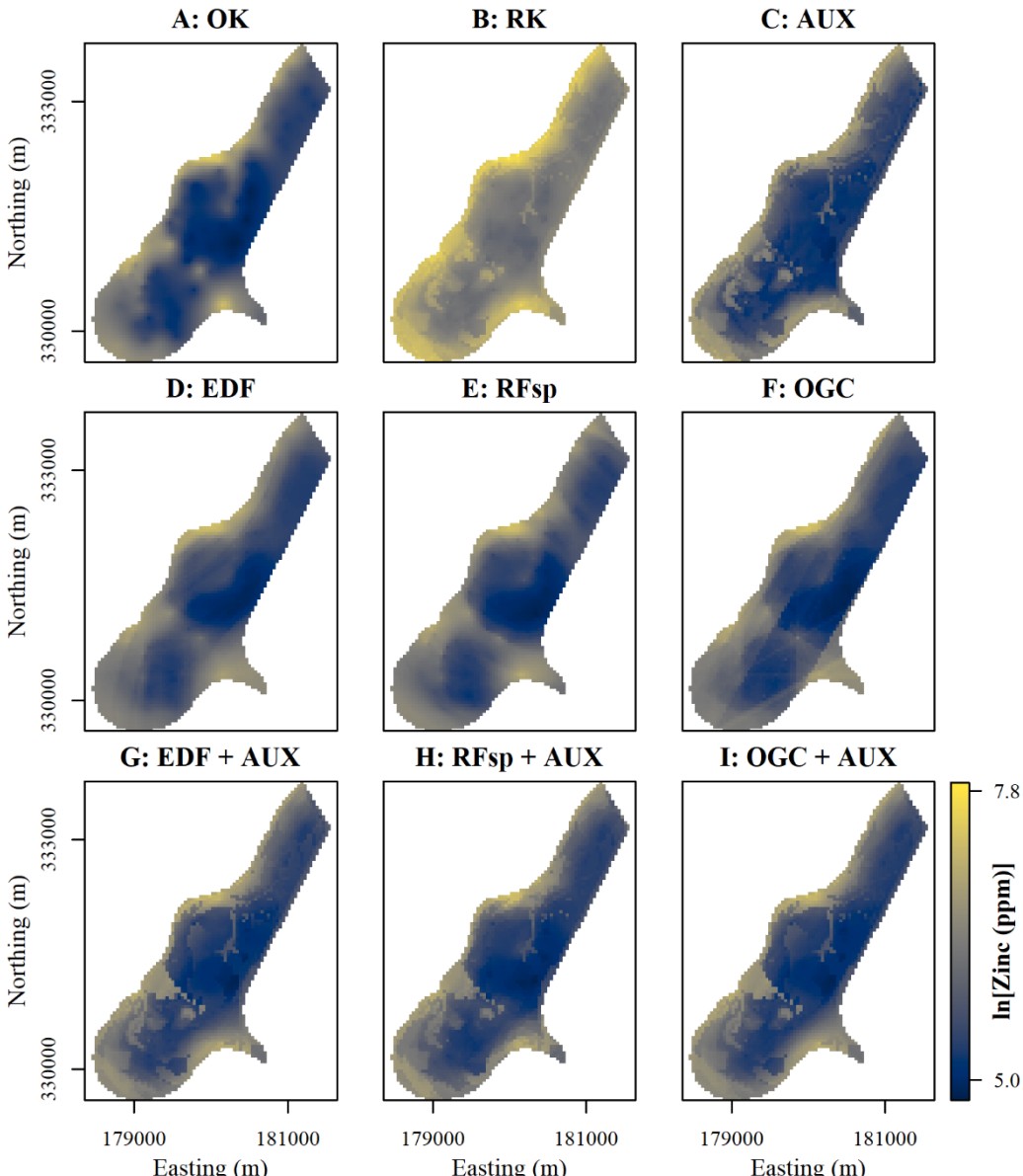

**Figure 9: Zinc contents predicted with each method for the *meuse* dataset. Easting and northing are Rijksdriehoek (RDH) (Netherlands topographical map) coordinates. A: Ordinary kriging (OK), B: Regression-kriging (RK), C: Auxiliary data (AUX), D: Euclidean distance fields (EDF), E: Distances to observations (RFsp), F: Oblique geographic coordinates (OGC), G: EDF**

**combined with AUX, H: RFsp combined with AUX, I: OGC combined with AUX.**

For the *eberg* dataset, predictions based only on auxiliary data showed a very noisy spatial pattern with many soil types occupying small incoherent areas (Figure 10C). The spatial patterns produced with OGC and especially EDF were much smoother and contained several large, rounded areas with little internal variation in soil types (Figure 10A and Figure 10B). The predictions obtained with the combined methods were similar to the spatial pattern obtained with only auxiliary data.

However, they were much smoother, as the soil types occupied mostly coherent areas. The effect for predictions of soil types therefore appears similar to the effect for numeric variables seen for Vindum and *meuse*.

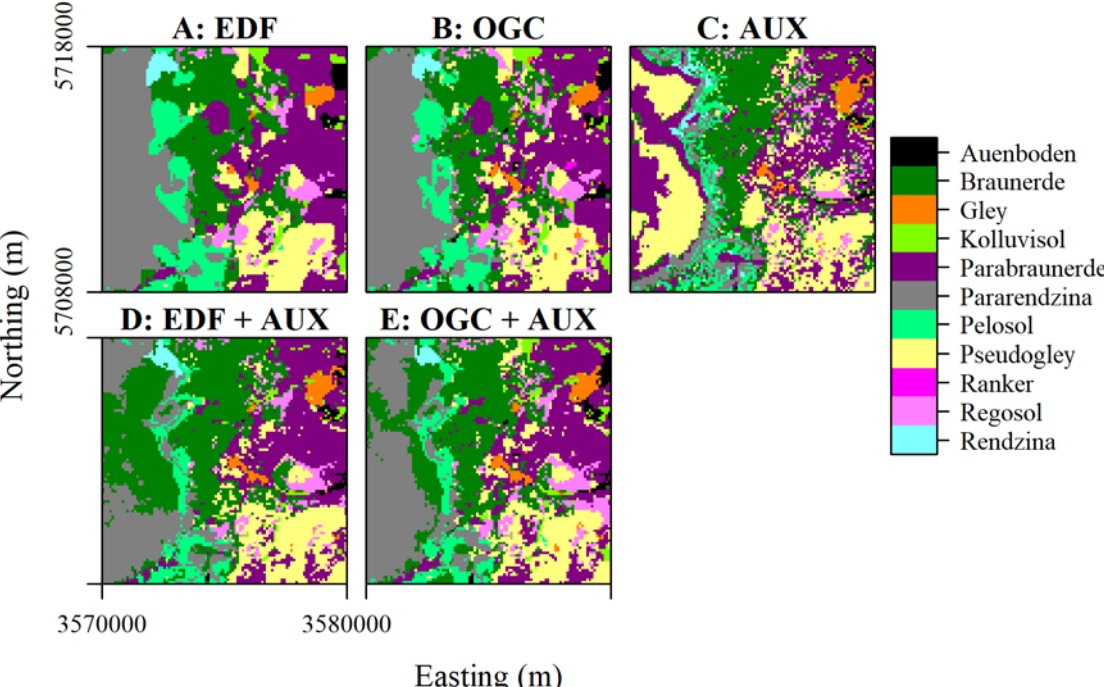

**Figure 10: Soil types predicted with each method for the *eberg* dataset. Easting and northing are coordinates according to DHDN / Gauss-Krueger zone 3 (German coordinate system). A: Euclidean distance fields (EDF). B: Oblique geographic coordinates (OGC). C. Auxiliary data (AUX). D: EDF combined with AUX. E: OGC combined with AUX.**

For the Swis rainfall dataset, OK produced a smooth, highly anisotropic prediction surface (Figure 11A). The prediction surfaces of EDF, RFsp and OGC also showed anisotropy, but they were generally smoother and more rounded. For example, with OK, some individual observations showed an effect on the prediction surface as elongated spots in the direction of the anisotropy. With the other three methods, a few individual observations showed an effect in the prediction surface, but their

effects are more rounded and less distinct. The predictions with EDF, RFsp and OGC therefore appear more general than the OK predictions. Moreover, the prediction surfaces of EDF, RFsp and OGC appear very similar.

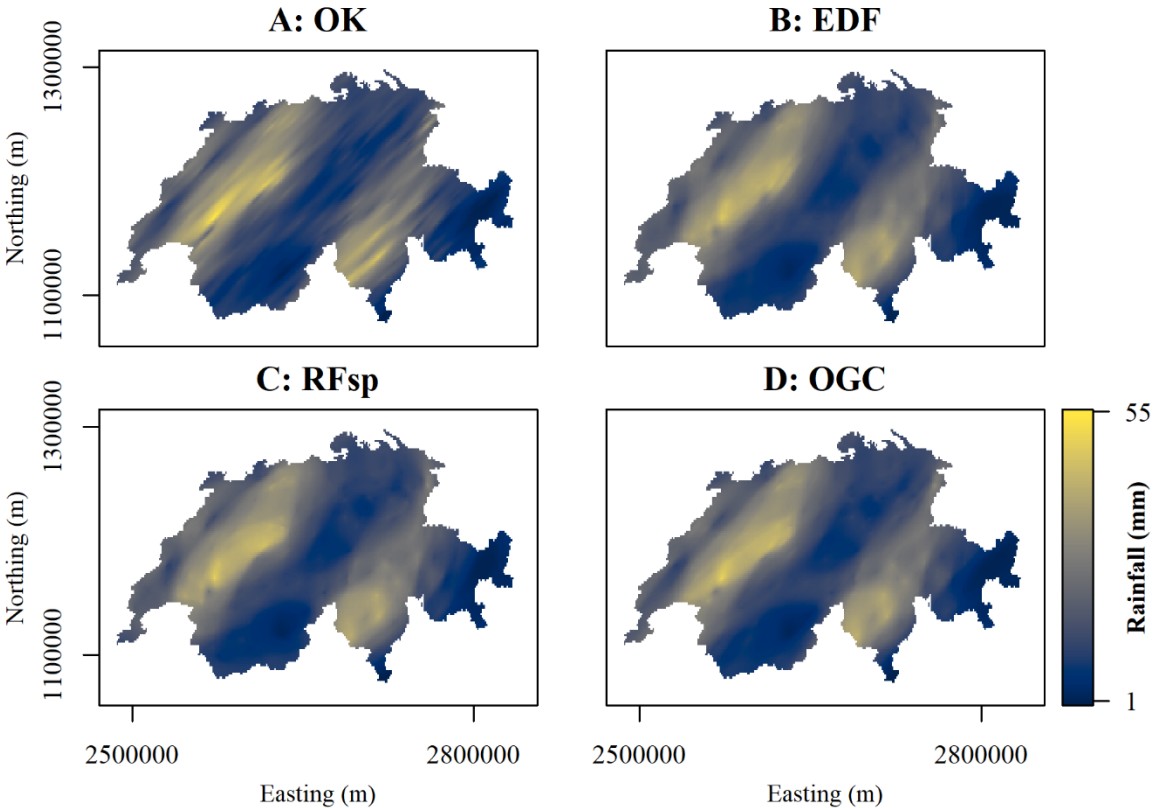

**Figure 11: Maps of rainfall on 1986-05-08 in Switzerland predicted with each method. Northing and easting are coordinates according to the Swiss coordinate system LV95. A: Ordinary kriging (OK). B: Euclidean distance fields (EDF). C: Distances to observations (RFsp). D: Oblique geographic coordinates (OGC).**

### 3.2.3 Residuals

For the Vindum dataset, the residuals of the SOM predictions had some degree of spatial dependence for all methods except kriging (Figure **12**). This finding contrasts with Hengl et al. (2018) who found that there was no spatial trend in the residuals of predictions with RFsp. EDF, RFsp and OGC used without auxiliary data had the most spatially dependent residuals. However, the residuals of the combined methods (EDF + AUX, RFsp + AUX and OGC + AUX) had less spatial dependence than the residuals of models based only on auxiliary data. OGC + AUX was the machine learning method with the least spatially dependent residuals, although the residuals still had more spatial dependence than kriging residuals.

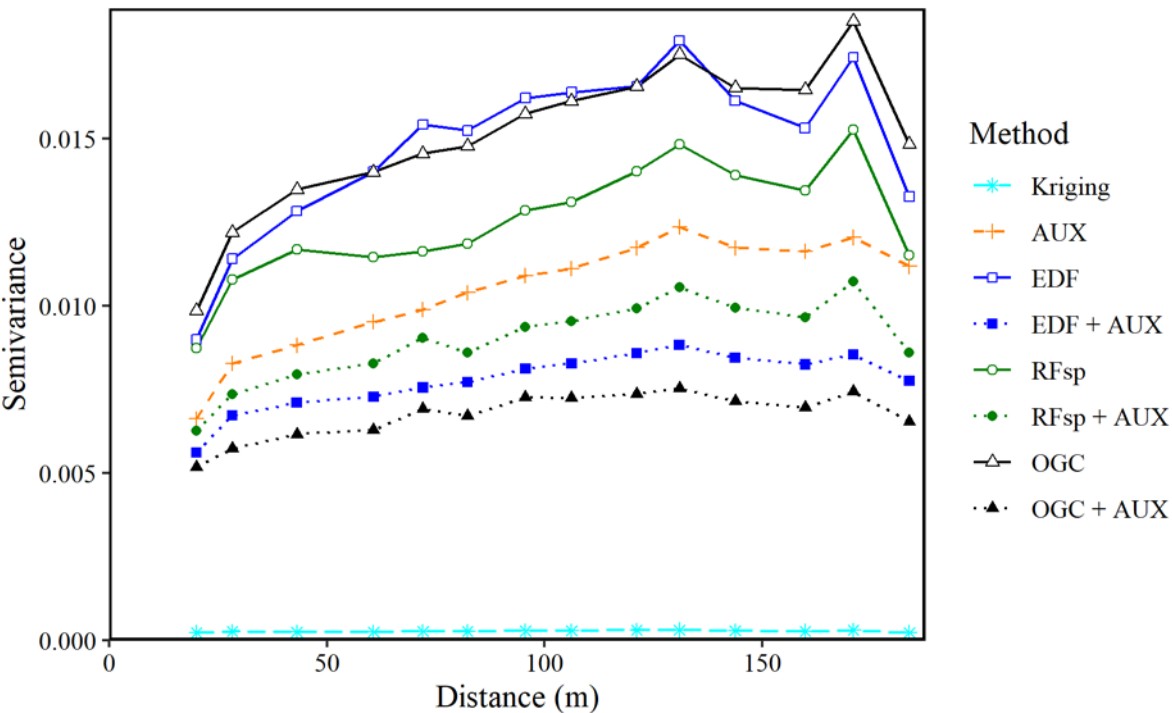

**Figure 12: Experimental variograms for the residuals of the SOM predictions made with each method for the Vindum dataset. The variograms use residuals from natural logarithmic-transformed SOM measurements and predictions. AUX: Auxiliary data. EDF: Euclidean distance fields. RFsp: Spatial Random Forest. OGC: Oblique geographic coordinates.**

### 3.3 Covariate importance

For the Vindum dataset, the most important covariate from the auxiliary data was the depth of sinks (Table 6). The most likely reason for its high importance is the presence of a large sink with very high SOM contents northwest of the middle of this study area (Figure 1). As sinks trap surface runoff, they often have wet conditions, which give rise to peat accumulation.

**Table 6: Covariate importance for the model using OGC in combination with auxiliary data for Vindum. The importance for OGC represents the sum of the importance of the coordinate rasters at 80 different angles.**

| Covariate | Importance (variance) |
| --- | --- |
| OGC | 2689 |
| Depth of sinks | 2003 |
| MRVBF | 476 |
| SAGA wetness index | 170 |
| Elevation | 166 |
| Valley depth | 157 |
| $EC_a$ | 123 |
| Slope gradient | 101 |
| Mid-slope position | 84 |
| NDVI | 76 |
| Plan curvature | 74 |
| SL | 64 |
| SAVI | 58 |
| Cos aspect | 44 |
| DVI | 42 |
| TWI | 38 |
| RVI | 34 |
| Flow accumulation | 32 |
| Sin aspect | 32 |
| Profile curvature | 21 |

When used in combination with the auxiliary data, the importance of the individual coordinate rasters varied from 0.6% to 3.1% of the importance of the depth of sinks, with mean value of 1.7%. The most important coordinate raster had $\theta = 0.48\pi$ (close to a north-south axis) and was the 12[th] most important covariate. The sum of the importance of the coordinate rasters was equal to 134.3% of the importance of the depth of sinks (Table 6). Therefore, with coordinate rasters at 80 different angles, the effect of the individual rasters on the predictions was subtle, but their combined effect was strong.

Figure 13 shows the importance of the coordinate rasters relative to $\theta$, in a way similar to a wind rose. The plots repeat the bars for $\theta \geq \pi$, as the importance for a given angle is directionless. For example, the importance of $\theta = 0$ (East) is equal to the importance of $\theta = \pi$ (West).

Without auxiliary data, the most important coordinate rasters had a general northwest to southeast angle (Figure 13). On the other hand, the coordinate rasters with angles between a north-south and a northeast-southwest axis had low importance. The

most likely reason for this pattern is the location of the sink with very high SOM contents to the northwest of the middle of this study area. This creates a large difference in the SOM contents of the northwestern and southeastern parts of the study area, giving large importance to covariates that can explain this difference. Additionally, the northwest side of the sink has a

very steep slope, creating a steep gradient in SOM contents in this direction. A stable variogram showed anisotropy along a north-north-east to south-south-west axis ($\theta = 0.34\pi$) with a major range of 136 m and a minor range of 118 m. The direction of the anisotropy therefore coincided with the direction of the least important coordinate rasters.

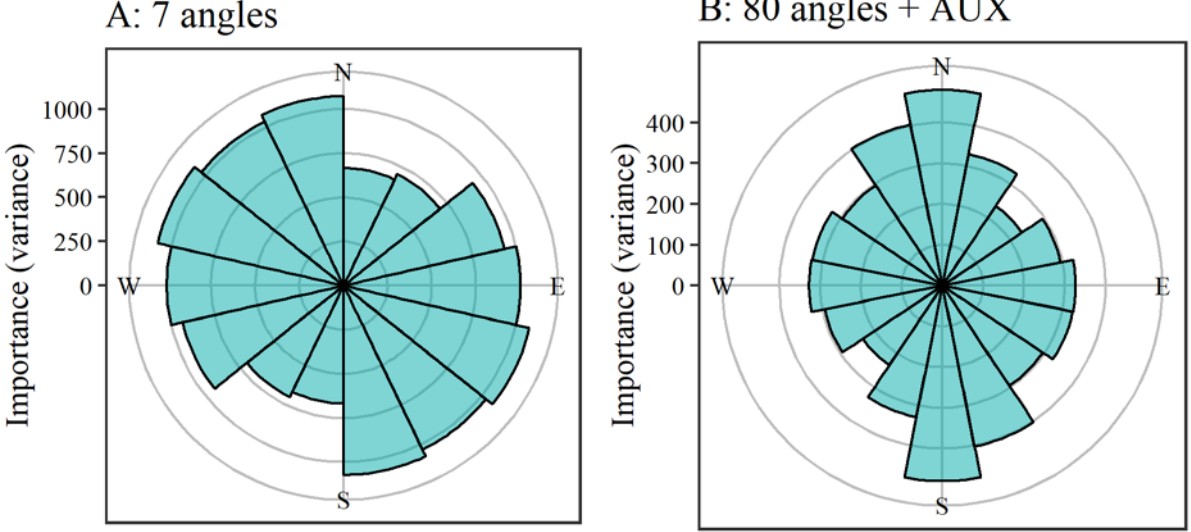

**Figure 13: Covariate importance of the coordinate rasters at various angles for Vindum. A: Importance of coordinate rasters at seven different angles. B: Importance of coordinate rasters at 80 different angles used with auxiliary data (importance of auxiliary data not shown). The sizes of the bars show the importance of the coordinate rasters at a given angle. Bars in B show the sum of the importance for coordinate rasters aggregated into $0.125\pi$ intervals.**

On the other hand, with OGC in combination with auxiliary data, the most important coordinate rasters had tilt angles close to a north-south axis ($\theta = 0.5\pi$). At the same time, the least important coordinate rasters had tilt angles close to a northeast-southwest axis ($\theta = 0.25\pi$). The residuals from the predictions with only auxiliary data also displayed a degree of anisotropy. A stable variogram showed anisotropy along a northeast to southwest axis ($\theta = 0.21\pi$), with a major range of 52 m and a minor range of 38 m. Again, the angle of the anisotropy coincided with the angle of the least important coordinate rasters. The spatial pattern of the residuals therefore differed from the spatial pattern of the SOM contents in the Vindum study area. Apparently, there are unaccounted-for processes decreasing the spatial variation along a northeast-southwest axis relative to other angles.

A possible cause of the anisotropy in the residuals may be the ploughing direction. The main ploughing direction in the Vindum study area is along an east-north-east to west-south-west axis ($\theta = 0.18\pi$). This angle is nearly parallel to the angle of the least important coordinate rasters (Figure 14). The ploughing direction, combined with the topography, has a large impact on soil movement, as ploughing displaces soil both along and across its direction (Lindstrom et al., 1990, De Alba, 2003, Heckrath et al., 2006). Most of the study area has the same ploughing direction, irrespective of the topography, resulting in up-, down- and cross-slope ploughing in various parts of the field. This creates in a complex pattern of soil

redistribution, which likely affects the SOM contents of the topsoil. As downslope soil movement is strongest in the

ploughing direction, variation in soil properties parallel to this direction is likely to be smaller than the variation

perpendicular to the ploughing direction. This corresponds to the low importance of coordinate rasters with angles close to

the ploughing direction. However, none of the auxiliary data accounted for the ploughing direction. This indicates that OGC

can add information on the most likely processes affecting soil properties in an area.

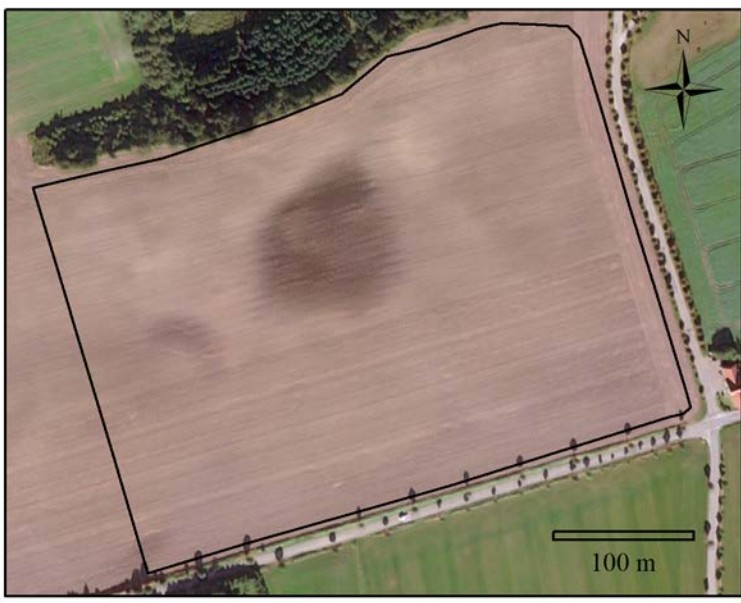


**Figure 14: Orthophoto of the study area from September 27, 2016 (Esri, 2019). Sources: Esri, DigitalGlobe, Earthstar Geographics, CNES/Airbus DS, GeoEye, USDA FSA, USGS, Aerogrid, IGN, IGP, and the GIS User Community.**

**3.4 Choice of method**

At Vindum, the three most accurate methods were kriging, RFsp with auxiliary data and OGC with auxiliary data. For

*meuse*, OGC and EDF combined with auxiliary data were most accurate, and for *eberg,* OGC combined with auxiliary data

was most accurate. For the Swiss rainfall dataset, OGC was the most accurate method.

Although kriging was in most cases less accurate than other methods, some soil mappers would probably still choose it for

mapping soil properties, due to its computational efficiency and conceptual simplicity. However, aside from accuracy, an

advantage of methods based on machine learning lies in the fact that they provide larger amounts of information than

geostatistical models. Kriging in itself does not provide information on the processes that control spatial variation in soil

properties, but machine learning models can include covariates related to soil processes, providing information on the

processes that are most likely to affect the spatial distribution of a soil property.

With spatial approaches such as EDF, RFsp and OGC, researchers can incorporate feature space and geographic space in a

machine learning model. Of the previously used approaches, OGC is most similar to EDF, as it used the x- and y-

coordinates, and the distances to the corners of the study area resemble coordinates. On the other hand, RFsp is more similar

to geostatistical models, as it relies on distances between observations. However, this similarity comes at the cost of calculating a large number of distance rasters.

One advantage of using spatially explicit covariates (EDF, RFsp or OGC) is that researchers can interpret local and spatial effects at once. In this regard, OGC has an advantage over EDF and RFsp, as it is clear what the coordinate rasters represent. It is less clear how researchers should interpret distances to the corners of the study area or the distance to a specific observation. We have also shown that it is straightforward to illustrate covariate importance for OGC.

Furthermore, an advantage of OGC relative to RFsp is that OGC required fewer covariates to achieve the same accuracy. In fact, without auxiliary data, OGC achieved a higher accuracy with a smaller number of covariates for the data sets Vindum, *meuse* and Swiss rainfall. This demonstrates a clear advantage of OGC, as it is possible to adjust the number of coordinate rasters. EDF and RFsp do not presently have similar options.

We will stress that, as a rule, soil mappers should not use machine learning models relying only on spatial trends, as EDF, RFsp and OGC all yielded lower accuracies without auxiliary data for the soil datasets (Vindum, *meuse* and *eberg*). Moreover, surprisingly, these methods had the most spatially autocorrelated residuals for the Vindum dataset, although they relied exclusively on spatial trends. The results therefore suggest that soil mappers should primarily use these methods in combination with auxiliary data, and not on their own. If no auxiliary data are available, kriging is likely to be a better option. However, results from the Swiss rainfall dataset show that, for other spatial problems, auxiliary data may be unnecessary.

**4 Conclusions**

We have shown in this study that the use of oblique geographic coordinates (OGC) is a reliable method for integrating auxiliary data with spatial trends for modelling and mapping soil properties. In most cases, the method eliminated the orthogonal artefacts that arise from the use of x- and y-coordinates and achieved higher accuracies than maps created with only two coordinate rasters. However, for *meuse*, without auxiliary data, OGC still produced abrupt angular artefacts in the final map. Soil mappers should therefore combine OGC with auxiliary data, as this gives higher accuracies and spatial patterns with a higher degree of realism.

OGC is more interpretable than previous similar approaches, and more flexible, as it is possible to adjust the number of coordinate rasters. This should allow soil mappers to find a good compromise between accuracy and computational efficiency for mapping soil properties, as the optimal number of coordinate rasters may vary depending on the study area and the soil property in question.

At this point, we have only tested the method for three soil datasets and one meteorological dataset. It will therefore be highly relevant to test the method for other soil properties and areas. It will especially be relevant to test the method in larger, less densely sampled areas. Previous studies have shown that machine learning is likely to provide higher accuracies in such areas (Zhang et al., 2008, Greve et al., 2010, Keskin et al., 2019), and it will be relevant to test if this is also the case for

oblique geographic coordinates. . Results from the Vindum and the Swiss rainfall datasets also suggest that the method can be useful for mapping variables with anisotropic spatial distributions, and it will therefore be relevant to test it on datasets

with a high degree of anisotropy. Lastly, one should note that we carried out this study for relatively small areas using "flat" coordinate systems. Using OGC for larger areas and other coordinate systems may require alterations to the method.

We call upon researchers within digital soil mapping to aid us in testing oblique geographic coordinates as covariates for additional datasets, and we have therefore made the function for generating oblique geographic coordinates available as an R package. Moreover, to allow other researchers to test methods on the Vindum dataset, we have made it available as well as

part of the same package.

## 5 Code and data availability

The function for generating oblique geographic coordinates is available as an R package at https://bitbucket.org/abmoeller/ogc/src/master/rPackage/OGC/. The package also contains the SOM observations and auxiliary data from the Vindum dataset.

Furthermore, we have made the R code used in this study available in a public repository at http://dx.doi.org/10.5281/zenodo.3820068.

## 6 Appendix A: Methods and results for additional datasets

### 6.1 Methods

#### 6.1.1 meuse

We mapped zinc contents for the *meuse* dataset (155 points). The *meuse* dataset contains covariates including the flooding frequency and the distance to the river. We added two covariates in the form of a digital elevation model (DEM, https://www.ahn.nl/) and surface water occurrence (Pekel et al., 2016). We converted the categorical raster of flooding frequency to indicator variables and transformed all the covariates to principal components. This resulted in six principal components.

We tested all the methods applied to the Vindum dataset, with the addition of regression-kriging (RK). We used Random Forest models trained on the auxiliary data for regression and then kriged the residuals using the function *krige.conv* from the R package *geoR* (Ribeiro Jr et al., 2020). As for the Vindum dataset, we tested each method with 100 repeated splits into training (75%) and test (25%) data. For each split, we calculated Pearson's $R^2$, RMSE and ccc. We carried out pairwise t-test on the accuracies obtained with each method in order to asses if the differences between their accuracies were statistically

significant. We also produced maps with each of the nine methods in order to compare results.

### 6.1.2 eberg

We mapped soil types for the *eberg* dataset. The *eberg* dataset contains 3,670 soil observations. We removed points outside the coverage of the covariates and points without a soil type classification. Furthermore, we removed the soil types "Moor" and "HMoor", as only one observation was available for each soil type. This reduced the dataset to 2,552 observations.

The *eberg* dataset contains covariates including the parent material, a DEM, the SAGA GIS topographic wetness index and the Thermal Infrared reflectance from satellite imagery. We converted the parent material classes to indicators and converted all covariates to principal components. This resulted in 11 principal components.

The dataset is highly clustered, which is likely to affect accuracy assessments, as some areas have much higher point densities than others. To counter this effect, we organized the data in 100 groups using k-means clustering on their

coordinates. We then produced 100 splits into training and test data based on these groups. In each split, the training data contained observations from 75 groups, and the test data contained observations from the remaining 25 groups.

As we aimed to predict a categorical variable, we did not use kriging. Furthermore, due to the large size of the dataset, we did not use RFsp, as this would require us to produce more than 2,000 raster layers with buffer distances. Hengl et al. (2018) avoided this by calculating only buffer distances to each soil type. However, we did not choose this solution, as it would

create problems for accuracy assessment. If a raster layer contains distances to test observations as well as training observations, the result would be circular logic, invalidating the accuracy assessment. Buffer distances based only on the training data would be less problematic. However, as we used 100 repeated splits, this was not an option.

We therefore tested only five methods for the *eberg* dataset: Models based on auxiliary data (AUX), Euclidean distance fields (EDF), OGC, as well as EDF and OGC combined with auxiliary data.

Due to the large size of the dataset, model training was slower than for the other datasets. We therefore tuned a Random Forest model only once for each method and used the resulting parameterization for all 100 data splits. For each split, we calculated accuracy on the test data as the fraction of observations correctly predicted. We carried out pairwise t-tests on the accuracies obtained with each method in order to assess if the differences between their accuracies were statistically significant.

We produced maps of soil types with each of the five methods in order to compare results.

### 6.1.3 Swiss rainfall

The Swiss rainfall dataset contains 467 rainfall observations from Switzerland from May 8, 1986. We did not use any covariates for this dataset, and we therefore tested only purely spatial methods. We tested ordinary kriging with correction for anisotropy, EDF, RFsp and OGC. As for the Vindum dataset, we tested each method with 100 repeated splits into

training data (75%) and test data (25%). For each split, we calculated Pearson's $R^2$, RMSE and ccc. We carried out pairwise t-tests on the accuracies obtained with each method in order to assess if the differences between their accuracies were statistically significant. Lastly, we produced maps of rainfall with each of the four methods in order to compare results.

## 6.2 Results

### 6.2.1 meuse

For the *meuse* dataset, the accuracy of OGC combined with auxiliary data was consistently higher than the accuracy of OGC without auxiliary data, irrespective of the accuracy metric and the number of coordinate rasters (Figure A1). The accuracy of OGC initially increased quickly with the number of coordinate rasters up to an optimum, after which there was no further improvement. For OGC + AUX, the increase in accuracy was more gradual, up to an optimum, after which accuracy deteriorated slightly. The optimal number of coordinate rasters without auxiliary data was six for RMSE and ccc and eight

for $R^2$. With auxiliary data, the optimal number of coordinate rasters was 11 for RMSE and 13 for $R^2$ and ccc. In the subsequent analysis, we used six coordinate rasters for OGC without auxiliary data and 11 coordinate rasters for OGC with auxiliary data.

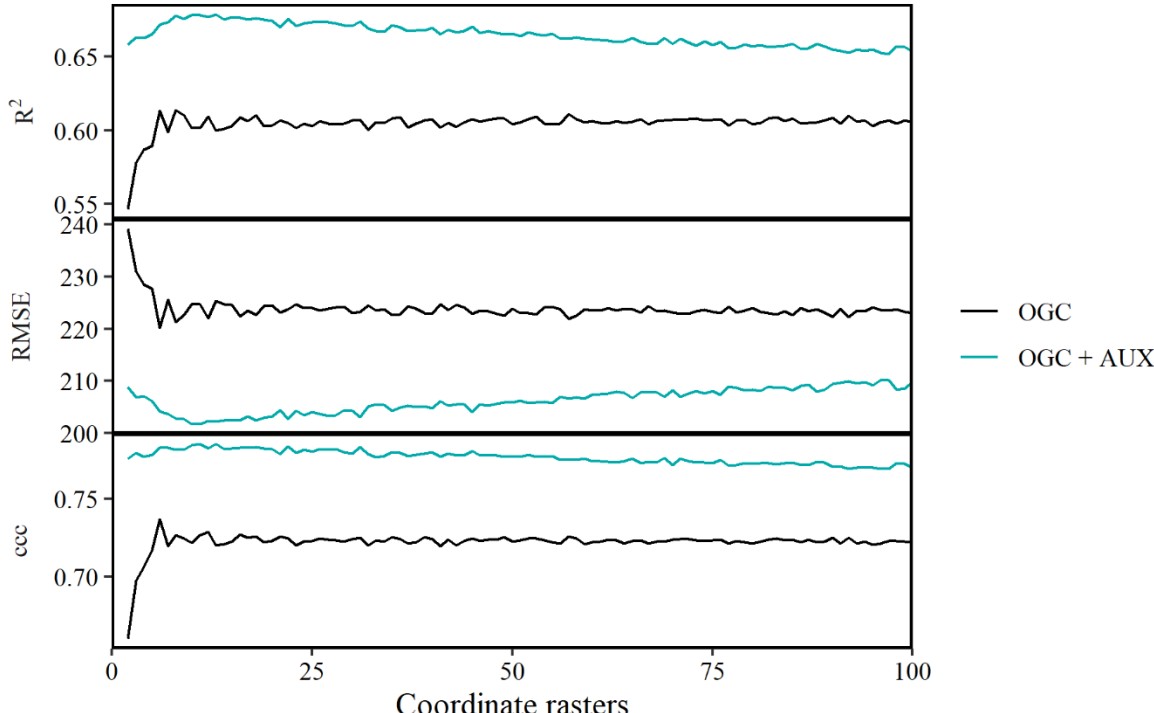

**Figure A1: Accuracy of predictions on the *meuse* dataset (zinc contents) versus the number of coordinate rasters with oblique**
**geographic coordinates, with and without auxiliary data. The values are averages obtained with 100 splits into training and test data.**

For the *meuse* dataset, as for the Vindum dataset, the differences between the accuracies of the methods was relatively small relative to the variation in accuracy between the test splits (Figure A2). Furthermore, most methods had long tails with lower accuracies.

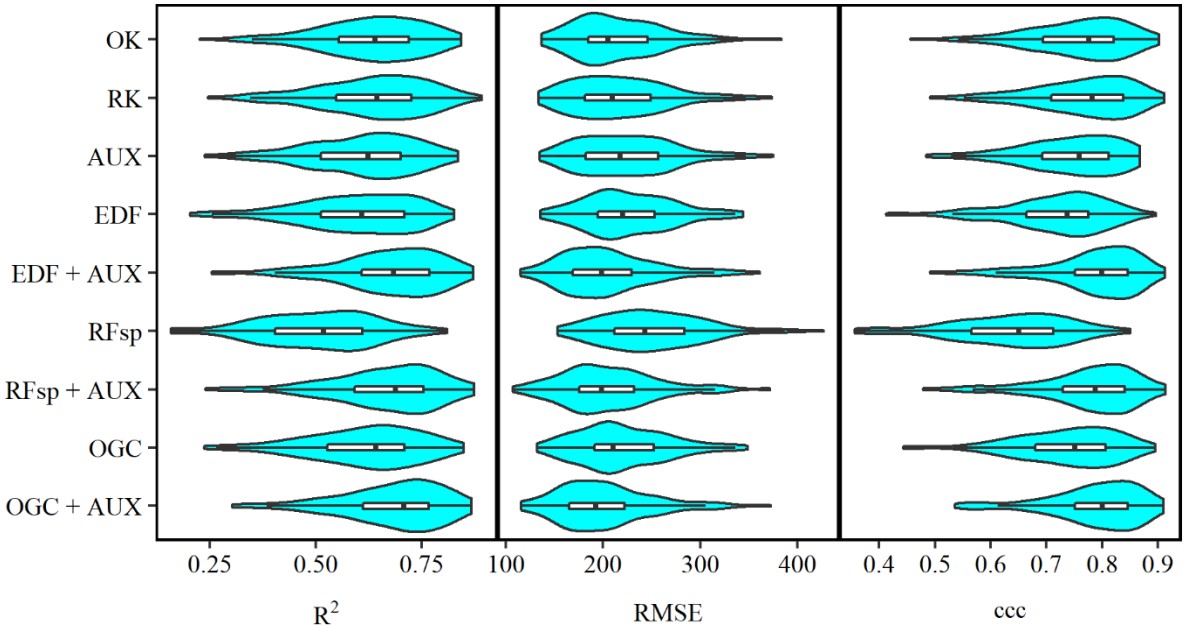

**Figure A2: Violin plots showing the accuracies obtained on the *meuse* dataset (zinc contents) with each method. The plots show values obtained with 100 splits into training and test datasets.**

### 6.2.2 eberg

For the *eberg* dataset, accuracy for OGC without auxiliary data first increased sharply up to five coordinate rasters. Past this

point, there was little improvement in accuracy, and some numbers of coordinate rasters produced sharp, irregular drops in accuracy (Figure A3). Combined with auxiliary data, the accuracy of OGC increased up to 22 coordinate rasters, after which it gradually declined. Without auxiliary data, the optimal number of coordinate rasters was 91. However, the highly irregular pattern of the accuracies did not justify any number past the initial increase, and we therefore used only five coordinate rasters. For OGC combined with auxiliary data, we used 22 coordinate rasters.

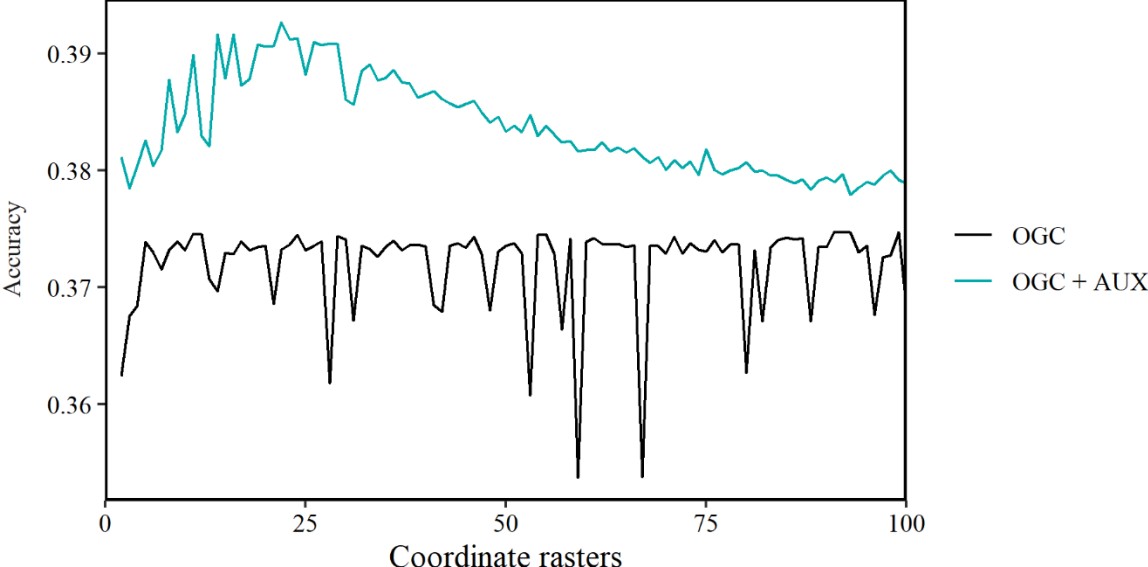


**Figure A3: Accuracy (percent of cases correctly predicted) of predictions on the *eberg* dataset versus the number of coordinate rasters with oblique geographic coordinates (OGCs), with and without auxiliary data (AUX). The values are averages obtained with 100 splits into training and test data.**

For the *eberg* dataset, as for the Vindum dataset, variation in accuracy between the splits into training and test data was in

most cases greater than variation between the methods (Figure A4). However, unlike the other datasets, the distributions of

the accuracies were mostly symmetric.

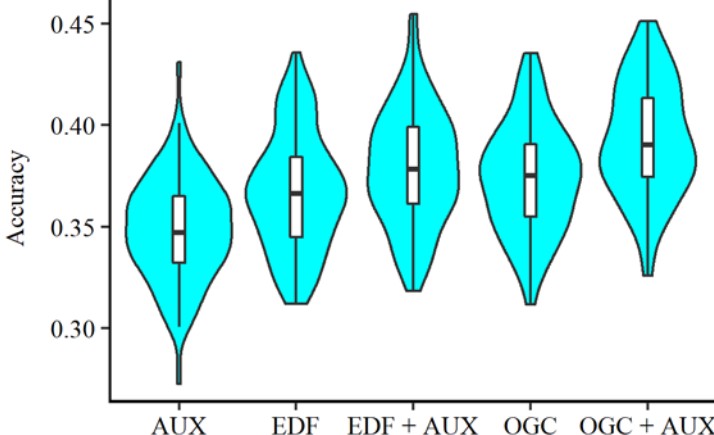

**Figure A4: Violin plot showing the accuracies obtained on the *eberg* dataset (percent correctly predicted) with each method. The plot shows values obtained with 100 splits into training and test datasets.**

 **6.2.3 Swiss rainfall**

For the Swiss rainfall dataset, the accuracy of OGC generally increased with the number of coordinate rasters (Figure A5). The increase in accuracy was steep at first, then gradual. For Pearson's $R^2$, the optimal number of coordinate rasters was 33, and for RMSE and ccc, it was 50. There was little change in accuracy past the optimal number of coordinate rasters.

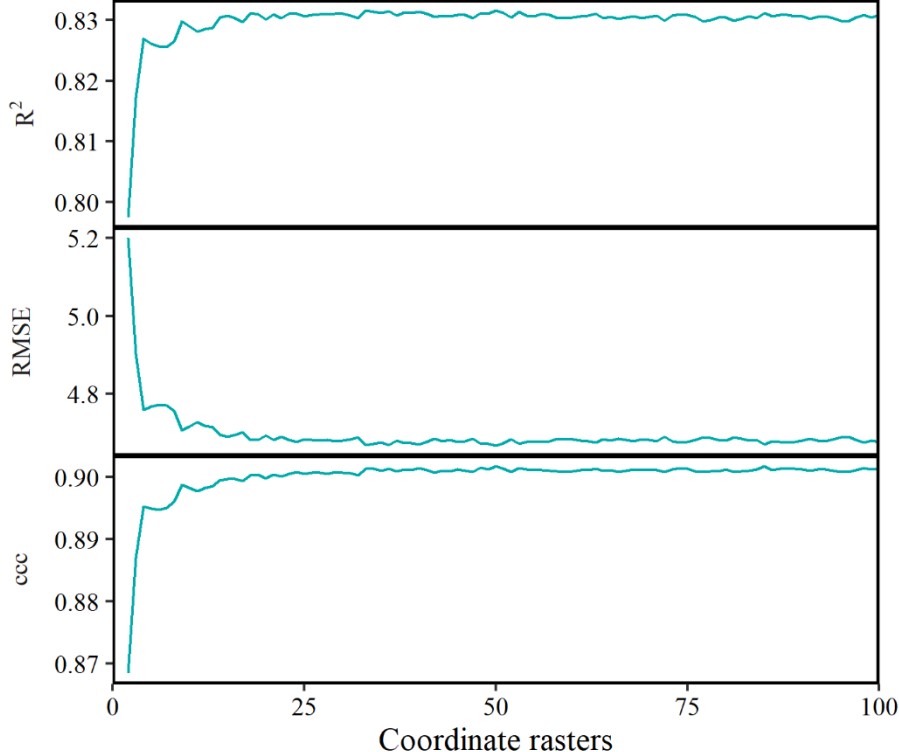

 **Figure A5: Accuracy of predictions on the Swiss rainfall dataset versus the number of rasters with oblique geographic coordinates. The values are averages obtained with 100 splits into training and test data.**

As for the other datasets, variation in accuracies on the Swiss rainfall dataset was greater between the splits into training and test data than between the methods (Figure A6). The distributions of RMSE were mostly symmetric, but the distributions of $R^2$ and ccc had long negative tails, as some splits yielded much lower accuracies than other splits.

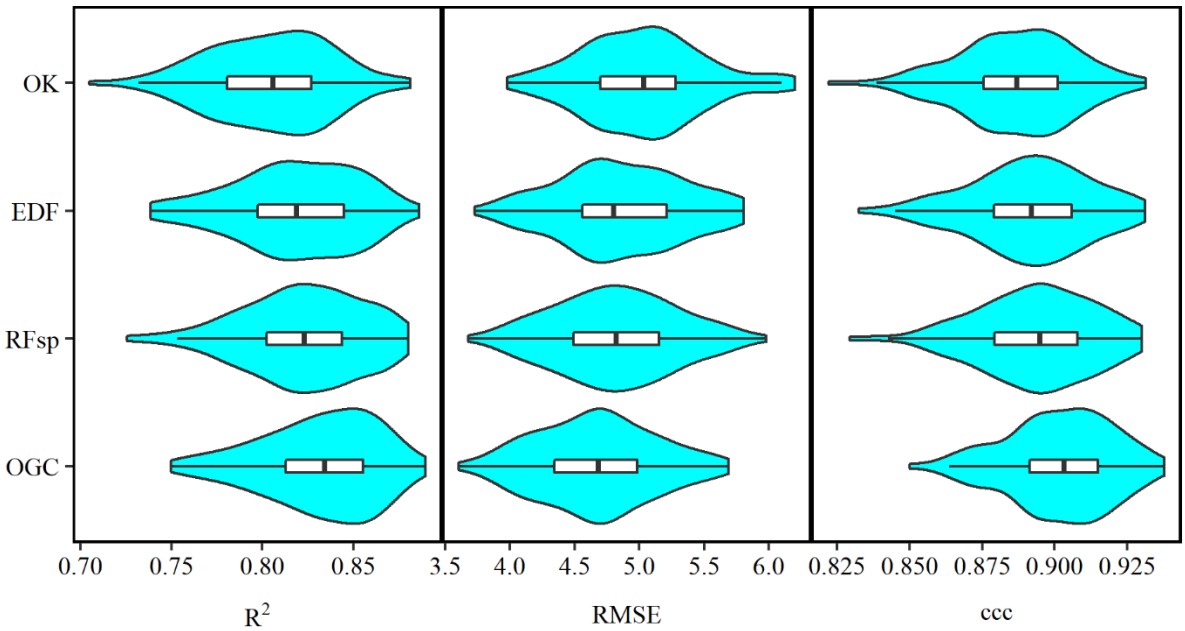


**Figure A6: Violin plots showing the accuracies obtained on the Swiss rainfal dataset with each method. The plots show values obtained with 100 splits into training and test datasets.**

## 7 Author contribution

Anders Bjørn Møller and Nastaran Pouladi prepared the data. Anders Bjørn Møller carried out the analyses and prepared the

manuscript with inputs from all co-authors.

## 8 Competing interests

The authors declare that they have no conflict of interest.

## 9 Acknowledgements

We are obliged to the two anonymous referees and to Dr. Alexandre Wadoux who provided vital feedback on the

manuscript. Their comments and advice have greatly improved the manuscript, and we give them our thanks.

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
