# Peer review of "Oblique geographic coordinates as covariates for digital soil mapping"

_SOIL, 2019_

## Short Comment (SC1) · 19 Nov 2019

This study tries to account for residuals spatial autocorrelation of a machine learning model by adding a set of pseudo-covariates. I have a few comments on the paper. I hope the authors find them useful and that it helps them to improve their manuscript. Overall, the study would benefit from a test of the method on several case studies, using different scales, different calibration sampling designs. A single case study at local scale and predicting a single soil property is in my opinion not enough to draw general conclusions.

About the methodology: 1) Any set of covariates with spatial pattern added to the original set of covariates may result in higher accuracy with a ML algorithm. This

is because ML algorithms can find relevant patterns even when the covariates are meaningless and not related to any soil forming process. The increase of accuracy that the authors obtain with the RF OGC + AUX model may well be obtained by adding any set of covariates with a spatial structure (see Fourcade et al., 2018).

2) Spatial autocorrelation in the raw data is not a problem per se and one should rather focus on remaining spatial autocorrelation on the residuals. I am strongly in favor of using only pedologically relevant covariates in a RF model. If the residuals of a model built using pedologically relevant covariates present autocorrelation, then one should consider making a map of the residuals because he may see a clear pattern of why this happens. The authors might then see that they are missing an important spatial process not included in the analysis. In this case one can add additional pedologically relevant covariates that could explain this pattern, and refit the model.

3) In case one made the previous step and admits that there is unexplained residual variation, one could consider using additional pseudo-covariates because there is no better proxy to explain the soil spatial variation. I stress here that these pseudo-covariates should not correlate with the pedological covariates because there would be redundancy (see next comment). In this case the pseudo-covariates should be covariates computed based on the remaining residuals. This would effectively tackle the problem of the residual autocorrelation and the authors would ensure that the pseudo-covariates do not interfere with the pedologically relevant covariates.

4) In this study, the authors include the set of pseudo-covariates with the set of pedologically relevant covariates. This is in my opinion very harmful because they can have pseudo-covariates which integrate over several of the pedologically relevant covariates, making them in some cases even better predictors. This is unrealistic and undesirable. This also makes the model less interpretable in terms of variable importance.

5) It is concluded that adding a set of pseudo-covariates effectively accounts for spatial autocorrelation in the data. This is clearly not the case as shown in Fig. 9 and admitted

by the authors at line 315 'the models built exclusively on spatial relationships had the most autocorrelated residuals.' The reason for this is that the covariates have a spatial pattern but are not related to the raw data and either to the residuals of the prediction made by a RF model. When the authors compared the sample variograms of kriging and RF residuals, it is visible that kriging do much better. The method would work if the sample variogram of RF OGC would be close to that of kriging. We can also see in Fig. 9 that the model with OGC covariates only have strong residual autocorrelation. The reduction in terms of residual autocorrelation in the OGC + AUX model is obtained by adding the pedologically relevant covariates. This is also a contradiction with the conclusion that OGC covariates account for the spatial autocorrelation.

6) Fig. 9 shows that there is still autocorrelation in the residuals of the RF model. This violates the assumption made in RF modelling, i.e. independence between the data points. Since this assumption is not satisfied, the calibrated RF model is potentially flawed. The authors have potentially missed important soil processes which could be added to the model as covariates. I would be interested to see a measure of the bias in the prediction.

Other considerations: Nugget to sill ratio should not be used to compare sample variograms, see Section 3.3. in https://doi.org/10.1016/j.catena.2013.09.006

Very surprised to read at Line 297 that the advantage of ML algorithms is their interpretability. I think the authors refer to the variable importance of the RF algorithm for the interpretability of the ML models. There is in my opinion a misunderstanding of the difference between ML and geo-statistical methods such as kriging. In ML you do not do inference and so you should not directly interpret the fitted model, or at least with caution. In geostatistics you can interpret because you make inference on the process that generated the data.

ML are also mostly black boxes. For example, is it impossible to interpret all the trees in a RF model, or all the neurons in a neural network model. This is in consequence

not justified to claim that ML algorithms have the advantage to be interpretable.

L 305: I would disagree with this conclusion; this would need to be justified by the literature or comparison between different case studies.

L 313: It is quite high accuracy a minimum CCC = 0.83.

L. 315. The authors have contradictory statements in the last paragraph of the Discussion. The last sentence is not very clear. Dealing with spatial data, which are auto correlated, a spatial methods is always needed otherwise you miss an important process and the fitted model is probably flawed because of the i.i.d assumption of the errors.

How did the authors compute the R2? A R2 can either indicate the closeness of the predicted values to the fitted regression line or the proportion of variance explained by the predictors. Authors should check that the R-square was computed against the 1:1 line and not against the fitted linear regression between observed and predicted, see https://doi.org/10.5194/soil-4-1-2018, Section 3.8 where the authors called it a skill score.

Impact of the sampling design is not considered. A spatial coverage design is very poor for random forest, while it is very efficient for kriging (assuming the variogram parameters are known). You should also consider that the sampling designs affect greatly the way the sample variograms are computed.

How did the authors compute the sample variograms? The authors gave no information about it.

It seems that the sample variogram for ordinary kriging is not at the same scale. It is either a much better model or the authors did not back-transformed the log-transformed observations. The authors mentioned that they log-transformed the observations prior to variogram fitting, it is not clear whether they also did it for the RF model.

---

## Author Comment (AC1) · 9 Dec 2019

**Reply to comments by Dr. Alexandre Wadoux on our manuscript "Oblique geographic coordinates as covariates for digital soil mapping"**

5    Anders Bjørn Møller[1], Amélie Marie Beucher[1], Nastaran Pouladi[1], Mogens Humlekrog Greve[1]

[1]Department of Agroecology, Aarhus University, Tjele, 8830, Denmark

Correspondence to: Anders Bjørn Møller (anbm@agro.au.dk)

**General reply**

10    We thank Dr. Alexandre Wadoux for the insightful comments on our manuscript "Oblique geographic coordinates as covariates for digital soil mapping" (Møller et al., 2019, Wadoux, 2019). We have found the comments very helpful in improving the manuscript, and we would like to give our replies to the comments.

We will start with a general reply to the commenter's use of "pseudocovariates" as a label for
15    oblique geographic coordinates. We see this label as misplaced. We believe the term "pseudocovariates" is only appropriate for covariates, which are clearly unsuited for the purpose, and this is not the case for oblique geographic coordinates.

Notable examples of pseudocovariates in the statistical literature have included randomly generated covariates for testing variable selection (Wu et al., 2007, Sandri and Zuccolotto,
20    2008, Sandri and Zuccolotto, 2009, Ghosal et al., 2019). In the mapping literature, recent studies have used pictures projected in geographic space as cautionary tales (Fourcade et al., 2018, Wadoux et al., 2019). The commenter correctly asserts that pseudocovariates with a spatial pattern can predict properties in geographic space with moderate success. However, we do not believe this to mean that researchers should disregard covariates that explicitly
25    account for spatial position.

In fact, the digital soil mapping literature has a rich number of studies, which have included spatial position as a covariate. The *scorpan* approach to digital soil mapping presented by McBratney et al. (2003) explicitly includes spatial position as a component. Although most studies in the review include spatial position through kriging or regression-kriging, the
30    authors are open to the use of covariates to account for spatial position. We quote:

"As was discussed in Section 2, soil can be predicted from spatial coordinates alone. […] This may indeed reflect some other environmental variable such as climate, and because of this it can be argued that n is not really a factor, but simply putting the coordinates is a simple way to ensure that spatial trends not included in the other environmental variables are not

35  missed. Therefore, n could also be described by some linear or nonlinear (nonaffine) transformation of the original spatial coordinates," (McBratney et al., 2003).

Oblique geographic coordinates represent such a transformation of the spatial coordinates. As one may expect from the previous reference, several studies have included x- and y-coordinates as covariates (Poggio and Gimona, 2014, Nussbaum et al., 2018, Koch et al.,

40  2019, Lagacherie et al., 2019). Other studies have included spatial position in the form of distance-based covariates, for example using distances to the coastline (Holmes et al., 2015) or rivers (Rudiyanto et al., 2018).

Recently, studies have included additional distance-based covariates, including distances to the corners and middle of the study area (Behrens et al., 2018b), and distances to observations

45  (Hengl et al., 2018). We hope therefore to have demonstrated that the use of covariates to account for spatial position is a theoretically sound, well-established practice, which does not warrant the label "pseudocovariates". Using covariates to include spatial position in machine learning models is in itself not new. Oblique geographic coordinates are simply a new method for doing this, with some advantages over previous methods.

50  In addition to this general reply, we would like to address the specific comments in the following.

**Specific replies**

We structure our replies by first showing the comment in question, then our reply to the comment.

55  COMMENT

This study tries to account for residuals spatial autocorrelation of a machine learning model by adding a set of pseudo-covariates. I have a few comments on the paper. I hope the authors find them useful and that it helps them to improve their manuscript. Overall, the study would benefit from a test of the method on several case studies, using different scales, different

60  calibration sampling designs. A single case study at local scale and predicting a single soil property is in my opinion not enough to draw general conclusions.

REPLY

In our manuscript, we mainly aim to introduce oblique geographic coordinates as a concept and to demonstrate the method on a dataset. We find that in this case it yields good accuracies

65  and meaningful results. We agree that it would be advantageous to test the method on several datasets in order to conclude more generally. However, it would also dilute the focus of the manuscript, as we would not be able to report the results in as much detail as we do.

We will add that it is usual to use only a single dataset for introducing a new method, as several studies have used this approach (Grimm et al., 2008, Odgers et al., 2014, Padarian et

70  al., 2019). We admit that there are notable exceptions (Behrens et al., 2018a, Behrens et al., 2018b, Hengl et al., 2018), but we still assert that our chosen approach is not problematic.

We agree that it is important to test oblique geographic coordinates on additional datasets, and we plan to do this in the future. This is also part of our rationale to share our code, as this will allow other researchers to test the method on their own datasets. This, we hope, will allow a more thorough assessment of the capabilities of the method.

**COMMENT**

About the methodology:1) Any set of covariates with spatial pattern added to the original set of covariates may result in higher accuracy with a ML algorithm. This is because ML algorithms can find relevant patterns even when the covariates are meaningless and not related to any soil forming process. The increase of accuracy that the authors obtain with the RF OGC + AUX model may well be obtained by adding any set of covariates with a spatial structure (see Fourcade et al., 2018).

**REPLY**

We concur that it would probably be possible to obtain the accuracies obtained with OGC + AUX with other (but not just any) sets of covariates. For example, RFsp + AUX achieves similar accuracies, although with a larger number of covariates. However, we will also remind the commenter that OGC do not simply have spatial structure – they have only spatial structure and nothing more. As we have already stated in our general reply, using covariates to account for spatial position is a well-established practice. OGC account for spatial position in a clear and systematic way, which is useful for decision tree algorithms and easily yields to interpretation.

**COMMENT**

2) Spatial autocorrelation in the raw data is not a problem per se and one should rather focus on remaining spatial autocorrelation on the residuals. I am strongly in favor of using only pedologically relevant covariates in a RF model. If the residuals of a model built using pedologically relevant covariates present autocorrelation, then one should consider making a map of the residuals because he may see a clear pattern of why this happens. The authors might then see that they are missing an important spatial process not included in the analysis. In this case one can add additional pedologically relevant covariates that could explain this pattern, and refit the model.

**REPLY**

We agree that it is important to use pedologically relevant covariates in machine learning models when mapping soil properties. We do not intend OGC to be used on their own, but in combination with auxiliary data of this form. As we hope to have demonstrated in our general reply, several studies have used spatially explicit covariates in combination with the other six components of the scorpan concept for digital soil mapping. Other studies have accounted for spatial autocorrelation in the residuals by means of regression-kriging, another well-established practice.

The commenter's dedication to purely pedologically relevant covariates has merit. However, due to the complexity of soil-forming processes, the hunt for a set of covariates that perfectly explain spatial variation in soil properties, is in many cases likely to be fruitless.

**COMMENT**

3) In case one made the previous step and admits that there is unexplained residual variation, one could consider using additional pseudo-covariates because there is no better proxy to explain the soil spatial variation. I stress here that these pseudocovariates should not correlate with the pedological covariates because there would be redundancy (see next comment). In this case the pseudo-covariates should be covariates computed based on the remaining residuals. This would effectively tackle the problem of the residual autocorrelation and the authors would ensure that the pseudocovariates do not interfere with the pedologically relevant covariates.

**REPLY**

Redundancy is generally not a risk for decision tree models, as they simply choose the optimal covariate in each split (Breiman, 2001). See also our reply to the next comment.

Furthermore, we doubt if the approach, which the commenter suggests, would be useful. We are not sure how the commenter would create a covariate based on the residuals. However, the attempt would create a serious risk of circular logic, which could invalidate model fitting and the assessment of model accuracy. Models should be based on covariates, not vice versa.

**COMMENT**

4) In this study, the authors include the set of pseudo-covariates with the set of pedologically relevant covariates. This is in my opinion very harmful because they can have pseudo-covariates which integrate over several of the pedologically relevant covariates, making them in some cases even better predictors. This is unrealistic and undesirable. This also makes the model less interpretable in terms of variable importance.

**REPLY**

Firstly, we refer to our general reply. Secondly, we will state that we see the commenter's allegation of "harmfulness" as a misunderstanding. We see the integration of spatial and environmental covariates as one of the strengths of using oblique geographic coordinates. Firstly, it allows the machine learnings model to map complex processes characterized by spatial dependence as well as environmental effects (Behrens et al., 2018b). This has an advantage over regression-kriging, where separate, mostly incomparable models treat environmental and spatial effects.

The commenter fears a scenario where a coordinate raster gains a higher importance than environmental covariates in a model. If this were the case, it would indeed be a cause of worry, but not for the reasons stated by the commenter. If a coordinate raster gains a higher importance than an environmental covariate, it suggests that the pedological process represented by the environmental covariate is probably not highly relevant for the soil property in this specific area. Therefore, if all environmental covariates turn out to be less important than coordinate rasters, it would show that the environmental covariates did not adequately account for spatial variation in the soil property.

150    In our case, the most important coordinate raster was the 12th most important covariate. OGC only became the second most important covariate, when we summed their importance. This shows that spatial effects have a large influence on SOM in the study area. However, it also shows that the model did not discard environmental covariates when we included OGC. Instead, it successfully integrated the two sets of covariates and their combined effects.

155    COMMENT

5) It is concluded that adding a set of pseudo-covariates effectively accounts for spatial autocorrelation in the data. This is clearly not the case as shown in Fig. 9 and admitted by the authors at line 315 'the models built exclusively on spatial relationships had the most autocorrelated residuals.' The reason for this is that the covariates have a spatial pattern but

160    are not related to the raw data and either to the residuals of the prediction made by a RF model. When the authors compared the sample variograms of kriging and RF residuals, it is visible that kriging do much better. The method would work if the sample variogram of RF OGC would be close to that of kriging. We can also see in Fig. 9 that the model with OGC covariates only have strong residual autocorrelation. The reduction in terms of residual

165    autocorrelation in the OGC + AUX model is obtained by adding the pedologically relevant covariates. This is also a contradiction with the conclusion that OGC covariates account for the spatial autocorrelation.

REPLY

We never claim in the manuscript that oblique geographic coordinates fully account for

170    spatial autocorrelation in the data. This comment would be more helpful if the commenter provided the lines where we allegedly state this.

We once refer to Hengl et al. (2018), who found that RFsp fully accounted for spatial autocorrelation in the data, but it is quite clear from the sentence that we refer to results in another study, not our own results. Our own results contrast with this earlier finding, and we

175    will include a comment on this in the final paper.

Furthermore, the commenter appears to reverse the interpretation of Figure 9. We intend soil mappers to use OGC as an addition to environmental covariates, not on their own. The figure shows that the addition of OGC greatly reduces spatial autocorrelation in the residuals relative to the model relying only on environmental covariates. We mainly include OGC,

180    EDF and RFsp on their own to demonstrate more clearly the effects of these sets of covariates. We do not recommend that researchers use them on their own.

COMMENT

6) Fig. 9 shows that there is still autocorrelation in the residuals of the RF model. This violates the assumption made in RF modelling, i.e. independence between the data points.

185    Since this assumption is not satisfied, the calibrated RF model is potentially flawed. The authors have potentially missed important soil processes which could be added to the model as covariates. I would be interested to see a measure of the bias in the prediction.

**REPLY**

We believe that it is quite an overstatement to say that any Random Forest model with
spatially autocorrelated residuals is potentially "flawed". Such a conclusion would most
likely invalidate a very large portion of Random Forest models used in digital soil mapping.
However, we agree that it is not an optimal situation, and that it might be useful to add more
environmental covariates.

As per the commenter's request, we have calculated bias as mean error (ME) for each
method. We have based this calculation on residuals from models using all observations:

| Method | ME |
| --- | --- |
| Kriging | -0.011 |
| AUX | 0.040 |
| EDF | 0.041 |
| EDF + AUX | 0.042 |
| RFsp | 0.011 |
| RFsp + AUX | 0.028 |
| OGC | 0.029 |
| OGC + AUX | 0.036 |

The values show that kriging has lower bias than the other methods except RFsp, but all
methods have low bias.

**COMMENT**

Other considerations: Nugget to sill ratio should not be used to compare sample variograms,
see Section 3.3. in https://doi.org/10.1016/j.catena.2013.09.006

**REPLY**

We see the error. In the final paper, we will remove mentions of the nugget-to-sill ratio when
comparing the variograms.

**COMMENT**

Very surprised to read at Line 297 that the advantage of ML algorithms is their
interpretability.

I think the authors refer to the variable importance of the RF algorithm for the interpretability
of the ML models. There is in my opinion a misunderstanding of the difference between ML
and geo-statistical methods such as kriging. In ML you do not do inference and so you should
not directly interpret the fitted model, or at least with caution. In geostatistics you can
interpret because you make inference on the process that generated the data.

ML are also mostly black boxes. For example, is it impossible to interpret all the trees in a RF
model, or all the neurons in a neural network model. This is in consequence not justified to
claim that ML algorithms have the advantage to be interpretable.

REPLY

This comment is confusing. The commenter appears to assert that (1) machine learning models are not interpretable, but that, on the other hand, (2) geostatistical models are interpretable. The commenter seems to conflate interpretation and inference, but we believe that one should understand these two as separate terms.

Furthermore, it gives the impression of a contradiction when the commenter states that machine learning models should not include spatial relationships, but also states that geostatistical models are interpretable. Likewise, the statement that machine learning models are not interpretable contrasts with the commenter's insistence that they should only contain covariates that represent pedological processes. If spatial position matters, even to the point where a geostatistical model is exclusively interpretable, why should we not use it in a model? Moreover, if we cannot interpret a machine learning model, then why does it matter what sort of covariates we use?

In themselves, geostatistical models only inform us on the spatial structure of the data. We agree that this can be useful, but any sort of interpretation would rely almost exclusively on the user's knowledge of the target variable and the processes that affect it. On the other hand, machine learning models are potentially far more informative.

Researchers should exert caution when interpreting any form of statistical model, but we agree with the commenter that it is especially relevant for machine learning models. Machine learning models are more complex than geostatistical models, and their interpretation is therefore also more complex and requires a higher level of abstraction. Tools to interpret machine learning models include covariate importance, which we use, but other tools exist, for example partial dependency plots (Friedman, 2001). Irrespective of the tools that researchers use, it is important that they critically use their knowledge of soils and the study area as well as the machine learning algorithm.

We can see that our statement that geostatistical models and machine learning models differ in interpretability is misleading. In the final paper, we will change the phrasing to state that the difference lies in the information content provided by the models.

COMMENT

L 305: I would disagree with this conclusion; this would need to be justified by the literature or comparison between different case studies.

REPLY

We cannot see why the commenter would outright disagree with this conclusion, as the commenter also states that spatial coverage sampling favors kriging. However, we do see the need for justification from the literature. Several studies have shown that machine learning models using environmental covariates are more accurate than geostatistical models for large, less densely sampled areas, including Zhang et al. (2008), Greve et al. (2010) and Keskin et al. (2019). We will include these references in the final paper.

**COMMENT**

255     L 313: It is quite high accuracy a minimum CCC = 0.83.

**REPLY**

We agree. In the final paper we will rephrase this sentence: "as EDF, RFsp and OGC all yielded lower accuracies without auxiliary data".

**COMMENT**

260     L. 315. The authors have contradictory statements in the last paragraph of the Discussion.

**REPLY**

We do not see the contradiction, but we agree that the sentences are not quite clear enough. In the final paper, we will rephrase the last two sentences: "The results suggest that these methods should be used in combination with auxiliary data, but not on their own. If no

265     auxiliary data are available, kriging is a better option."

**COMMENT**

The last sentence is not very clear. Dealing with spatial data, which are auto correlated, a spatial methods is always needed otherwise you miss an important process and the fitted model is probably flawed because of the i.i.d assumption of the errors.

270     ## REPLY

We agree on the lack of clarity. Please see our reply to the previous comment.

**COMMENT**

How did the authors compute the R2? A R2 can either indicate the closeness of the predicted values to the fitted regression line or the proportion of variance explained by the predictors.

275     Authors should check that the R-square was computed against the 1:1 line and not against the fitted linear regression between observed and predicted, see https://doi.org/10.5194/soil-4-1-2018, Section 3.8 where the authors called it a skill score.

**REPLY**

We used Peason's R2, this is, closeness to a fitted regression line. We see that we did not

280     include this information in the manuscript, and we will make sure to include it in the final paper.

We will not change the way we calculate R2, as Pearson's R2 indicates if the predictions have the same trend as the observations, which we believe is relevant in itself. We rely on several accuracy metrics, including also RMSE and CCC. CCC gives information on

285     closeness to a 1:1 line, which the commenter requests. Furthermore, the skill score, to which the commenter refers, uses on the mean square error (MSE) of the predictions, and the variance in the dataset. It is very useful for comparing accuracies across different regression problems. However, for any single regression problem, as in our study, the variance in the dataset will be constant, and variation in the skill score will depend only on variation in MSE.

290     As we already provide RMSE, this information would be redundant.

**COMMENT**

Impact of the sampling design is not considered. A spatial coverage design is very poor for random forest, while it is very efficient for kriging (assuming the variogram parameters are known). You should also consider that the sampling designs affect greatly the way the sample variograms are computed.

**REPLY**

We agree that the sampling design favors kriging. In fact, we already state in the manuscript that an earlier study in the same area (Pouladi et al., 2019) found that kriging yielded higher accuracies than machine learning models. It is therefore quite remarkable that OGC + AUX and RFsp + AUX allow Random Forest models to achieve accuracies on par with kriging.

**COMMENT**

How did the authors compute the sample variograms? The authors gave no information about it.

**REPLY**

Firstly, we produced maps with each method using all observations. Secondly, we converted both observations and predictions to natural logarithmic scale. We then subtracted the predictions from the observations and calculated variograms for these residuals. For this purpose, we used the function 'variogram' from the R package 'gstat' with its default parameters. We will include this information in the final paper.

Furthermore, we have discovered an error in our code, which caused us to use only 75% of the observations when calculating the variograms. We have therefore recalculated the variograms using all observations and produced a new version of Figure 9. We will include this updated figure in the final paper:

[Figure]

 COMMENT

It seems that the sample variogram for ordinary kriging is not at the same scale. It is either a much better model or the authors did not back-transformed the log-transformed observations. The authors mentioned that they log-transformed the observations prior to variogram fitting, it is not clear whether they also did it for the RF model.

320 REPLY

The variograms are all on the same scale. Kriging has smaller residuals than the other methods, as the variogram had a very small nugget, but we do not believe that this shows it to be a "better" model. For example, with inverse distance weighting interpolation, the residuals would be zero, but it would not necessarily by a very good model. We will also point out to
325 the commenter that the residuals for OGC + AUX show nearly no trend. So the residuals are larger, but they have very little spatial autocorrelation.

**References**

Behrens, T., Schmidt, K., MacMillan, R.A. and Rossel, R.A.V. Multi-scale digital soil mapping with deep learning. Sci. Rep. 8(1), 15244. http://dx.doi.org/10.1038/s41598-018-33516-6, 2018a.

Behrens, T., Schmidt, K., Viscarra Rossel, R., Gries, P., Scholten, T. and MacMillan, R. Spatial modelling with Euclidean distance fields and machine learning. Eur. J. Soil Sci. 69(5), 757-770. http://dx.doi.org/10.1111/ejss.12687, 2018b.

Breiman, L. Statistical modeling: The two cultures. Stat. Sci. 16(3), 199-215. http://dx.doi.org/10.1214/ss/1009213726, 2001.

Fourcade, Y., Besnard, A.G. and Secondi, J. Paintings predict the distribution of species, or the challenge of selecting environmental predictors and evaluation statistics. Glob. Ecol. Biogeogr. 27(2), 245-256. http://dx.doi.org/10.1111/geb.12684, 2018.

Friedman, J.H. Greedy function approximation: A gradient boosting machine. Ann. Stat. 29(5), 1189-1232. http://dx.doi.org/10.1214/aos/1013203451, 2001.

Ghosal, R., Maity, A., Clark, T. and Longo, S.B. Variable Selection in Functional Linear Concurrent Regression. arXiv preprint arXiv:1904.08507. 2019.

Greve, M.H., Greve, M.B., Kheir, R.B., Bøcher, P.K., Larsen, R. and McCloy, K. Comparing Decision Tree Modeling and Indicator Kriging for Mapping the Extent of Organic Soils in Denmark, in: Boettinger, J.L., Howell, D.W., Moore, A.C., Hartemink, A.E. and Kienast-Brown, S. (Eds.), Digital Soil Mapping: Bridging Research, Environmental Application, and Operation. Springer Netherlands, Dordrecht, 267-280, 2010.

Grimm, R., Behrens, T., Märker, M. and Elsenbeer, H. Soil organic carbon concentrations and stocks on Barro Colorado Island - Digital soil mapping using Random Forests analysis. Geoderma 146(1-2), 102-113. http://dx.doi.org/10.1016/j.geoderma.2008.05.008, 2008.

Hengl, T., Nussbaum, M., Wright, M.N., Heuvelink, G.B. and Gräler, B. Random forest as a generic framework for predictive modeling of spatial and spatio-temporal variables. PeerJ 6, e5518. http://dx.doi.org/10.7717/peerj.5518, 2018.

Holmes, K.W., Griffin, E.A. and Odgers, N.P. Large-area spatial disaggregation of a mosaic of conventional soil maps: Evaluation over Western Australia. Soil Res. 53(8), 865. http://dx.doi.org/10.1071/sr14270, 2015.

Keskin, H., Grunwald, S. and Harris, W.G. Digital mapping of soil carbon fractions with machine learning. Geoderma 339, 40-58. http://dx.doi.org/10.1016/j.geoderma.2018.12.037, 2019.

Koch, J., Berger, H., Henriksen, H.J. and Sonnenborg, T.O. Modelling of the shallow water table at high spatial resolution using random forests. Hydrol. Earth Syst. Sci. 23(11), 4603-4619. http://dx.doi.org/10.5194/hess-23-4603-2019, 2019.

Lagacherie, P., Arrouays, D., Bourennane, H., Gomez, C., Martin, M. and Saby, N.P.A. How
365    far can the uncertainty on a Digital Soil Map be known?: A numerical experiment using
pseudo values of clay content obtained from Vis-SWIR hyperspectral imagery. Geoderma
337, 1320-1328. http://dx.doi.org/10.1016/j.geoderma.2018.08.024, 2019.

McBratney, A.B., Mendonça Santos, M.L. and Minasny, B. On digital soil mapping.
Geoderma 117(1-2), 3-52. http://dx.doi.org/10.1016/s0016-7061(03)00223-4, 2003.

370    Møller, A.B., Beucher, A.M., Pouladi, N. and Greve, M.H. Oblique geographic coordinates
as covariates for digital soil mapping. SOIL Discuss. http://dx.doi.org/10.5194/soil-2019-83,
2019.

Nussbaum, M., Spiess, K., Baltensweiler, A., Grob, U., Keller, A., Greiner, L., Schaepman,
M.E. and Papritz, A. Evaluation of digital soil mapping approaches with large sets of
375    environmental covariates. SOIL 4(1), 1-22. http://dx.doi.org/10.5194/soil-4-1-2018, 2018.

Odgers, N.P., Sun, W., McBratney, A.B., Minasny, B. and Clifford, D. Disaggregating and
harmonising soil map units through resampled classification trees. Geoderma 214-215, 91-
100. http://dx.doi.org/10.1016/j.geoderma.2013.09.024, 2014.

Padarian, J., Minasny, B. and McBratney, A.B. Using deep learning for digital soil mapping.
380    Soil 5(1), 79-89. http://dx.doi.org/10.5194/soil-5-79-2019, 2019.

Poggio, L. and Gimona, A. National scale 3D modelling of soil organic carbon stocks with
uncertainty propagation - An example from Scotland. Geoderma 232-234, 284-299.
http://dx.doi.org/10.1016/j.geoderma.2014.05.004, 2014.

Pouladi, N., Møller, A.B., Tabatabai, S. and Greve, M.H. Mapping soil organic matter
385    contents at field level with Cubist, Random Forest and kriging. Geoderma 342, 85-92.
http://dx.doi.org/10.1016/j.geoderma.2019.02.019, 2019.

Rudiyanto, Minasny, B., Setiawan, B.I., Saptomo, S.K. and McBratney, A.B. Open digital
mapping as a cost-effective method for mapping peat thickness and assessing the carbon
stock of tropical peatlands. Geoderma 313, 25-40.
390    http://dx.doi.org/10.1016/j.geoderma.2017.10.018, 2018.

Sandri, M. and Zuccolotto, P. A bias correction algorithm for the Gini variable importance
measure in classification trees. J. Comput. Graph. Stat. 17(3), 611-628.
http://dx.doi.org/10.1198/106186008x344522, 2008.

Sandri, M. and Zuccolotto, P. Analysis and correction of bias in Total Decrease in Node
395    Impurity measures for tree-based algorithms. Stat. Comput. 20(4), 393-407.
http://dx.doi.org/10.1007/s11222-009-9132-0, 2009.

Wadoux, A. Interactive comment on "Oblique geographic coordinates as covariates for
digital soil mapping" by Anders Bjørn Møller et al. SOIL Discuss.
http://dx.doi.org/10.5194/soil-2019-83-SC1, 2019.

400     Wadoux, A.M.J.C., Samuel-Rosa, A., Poggio, L. and Mulder, V.L. A note on knowledge
         discovery and machine learning in digital soil mapping. Eur. J. Soil Sci.
         http://dx.doi.org/10.1111/ejss.12909, 2019.

         Wu, Y.J., Boos, D.D. and Stefanski, L.A. Controlling variable selection by the addition of
         pseudovariables. J. Am. Stat. Assoc. 102(477), 235-243.
405     http://dx.doi.org/10.1198/016214506000000843, 2007.

         Zhang, X., Lin, F., Jiang, Y., Wang, K. and Wong, M.T. Assessing soil Cu content and
         anthropogenic influences using decision tree analysis. Environ. Pollut. 156(3), 1260-1267.
         http://dx.doi.org/10.1016/j.envpol.2008.03.009, 2008.

---

## Referee Comment (RC1) · Anonymous Referee #1 · 6 Jan 2020

The present paper is well written and structured. Moreover, the study aims to make a contribution to the field of DSM by providing a novel methodological framework based on the usage of coordinates. This is something that can be considered as rather 'out of the box thinking', because most attention in the international literature goes either to the use of advanced geostatistical methods (e.g. capturing the spatial autocorrelation through kriging) or external drift fitting based on '(environmental) co-variates' or a combination of both. Hence, the work certainly merit respect for its originality and the methodological framework seems to provide useful thoughts to be considered in future DSM-studies. However, I also see some shortcomings which should be addressed/considered in order to maximize its potential to be applied widely, and as such, I am looking forward receiving the authors replies on the associated comments

and suggestions presented below.

Major Comments: I believe that the main issue with this research is that it considers only one rather small field characterized by 1 remarkable / specific spatial structure as regards the variation of SOM (i.e. one spot/area with clearly higher values) in order to test the validity of the present new methodology, whereas the authors claim that the method will be highly useful for mapping soil properties in larger areas. Hence, I believe that the present methodology requires further testing by considering larger areas (e.g. catchment-regional scale) with more complex spatial patterns in SOM in order to prove the validity of the statements that have been made in this respect. Moreover, it would also be interesting to consider other key soil variables (besides SOM) to check whether the usages of oblique geographic coordinates as covariates could be seen as a universal DSM approach. In this context, I believe that using a national soil inventory database could be a good way forward. I may understand that this might not be possible in this study, but I still believe that this should be mentioned clearly (as a critical note) in the discussions (and maybe be picked up by the authors in future research).

When I have a look at the performance of the different mapping methods (as presented in the Violin plots in figure 7), it seems to me that your new OGC (+AUX) method only results in (very) small improvements as compared to some other (more commonly used) methods such as Kriging. Hence, I was wondering whether this improvement is statistically significant? And if this might still be the case when either (i) another field (characterized by a different spatial pattern), (ii) another soil variable or (iii) larger geographical extent are considered?

Minor Comments: I'm not too sure if it is entirely appropriate to use R2 as a measure to compare the different methods, because (i) a very high R2 value may also mean an 'overfit' and (ii) each method has it own degree of (model) complexity. Hence, I guess that it could be a good idea to take (also) another statistical measure into consideration that specifically aims to evaluate the methods' performance taking into account its complexity (in order to avoid overfitting)?

Figure 1 - Subpanel C: Showing hill shade is not enough to give the reader an insight into the topographical configuration of the field. Hence, I suggest adding contour lines.

---

## Referee Comment (RC2) · Anonymous Referee #2 · 31 Jan 2020

The manuscript "Oblique geographic coordinates as covariates for digital soil mapping" from Møller et al. presents a valuable contribution to integrate predictor information on spatial position into machine learning approaches for digital soil mapping. It, thereby, seeks to overcome the known problem of orthogonal artefacts sometimes introduced by the usage of xy-coordinates as covariates in recursive partitioning algorithms. While commonly applied covariates usually relate to site characteristics that approximate the soil forming factors, the inclusion of coordinates provides a chance to reflect further spatial patterns we are not necessarily aware of. The authors show that the usage of a multitude of oblique spatial coordinates reflects spatial anisotropy. Major spatial axes identified through predictor importance measures may then give a hint on the geographic direction of the underlying processes as the authors demonstrate. The

article compares the new approach (OGC) to existing approaches such as Euclidean distance fields (EDF) and spatial Random Forest (RFsp). The article is written using adequate language and it follows a clear structure. The figures are well prepared. Furthermore, it is a rare, but highly welcome choice of the authors to provide the R code of their approach.

While the authors very clearly demonstrate the power of their approach particularly due to the clear figures and the comparison to similar approaches, certain aspects would require reconsideration:

- I do not understand why the OGC+AUX approach is not directly compared to regression kriging, but to ordinary kriging. Ordinary kriging would require a stationary mean which is not given in this particular research setting. Accordingly, a regression model would first have to be fitted to model the trend from covariate data, while then spatial autocorrelation in the residuals will be accounted for by ordinary kriging of the residuals. While the regression model is fitted by random forest, this would also allow for direct comparability. The authors provide rather vague arguments against regression kriging (lines 27-32). - The data in this study display spatial autocorrelation. Specifically, a range of 139 m is mentioned. This is not surprising due to the high spatial data density. Furthermore, the authors mention a couple of processes that may have caused this spatial dependency. However, this aspect is not accounted for in the evaluation approach. 100 random splits 75/25 (training/ test set) make it very likely that spatially autocorrelated sampling points will end up in the test and training set for the majority of the 100 splits. As a consequence, the test sets are not independent of the training sets and will lead to overly optimistic error values. This aspect at least needs to be mentioned. Particularly, in the context of Figure 7.

The argumentation line of the introduction requires some improvement. Certain aspects need to be better clarified:

- The main advantage of OGC+AUX over using only XY+AUX is the high number of coordinates, as the usage of only two oblique coordinates would lead to similar artefacts as demonstrated in the results. - The usage of coordinates as predictors in a regression model differs from fitting a geostatistical model to the residuals of a regression model. The approach closest to fitting a semivariogram is RFsp, as it accounts for the distance between points. However, it comes at the cost of introducing a high number of covariates as the authors state, correctly. It is important that the authors also compare their approach to RFsp, but the difference in calculating a different set of coordinates and taking the distance between points into account should be explicitly mentioned. In contrast, OGC+AUX and EDF+AUX really follow a similar approach in calculating a set of different coordinates. OGC+AUX is demonstrated to be superior to EDF+AUX. - Overall, whether it is worse to make the effort of calculating a high number of oblique coordinates could only be decided while being compared to regression kriging.

There are a couple of statements that are problematic. Please consider rephrasing:

- lines 19-21 "...decision tree algorithms...are immune to correlated and redundant covariates". There are a couple of publications that show the contrary. - line 29 "By kriging the residuals...soil mappers have been able to reduce or remove spatial bias". We usually fit a geostatistical model to explain spatial autocorrelation not to remove spatial bias. Please also correct throughout the manuscript, e.g. lines 45/46. - line 47 "...methods are able to integrate spatial relationships..." I am not convinced that by the mere consideration of coordinates we account for spatial relationships, leave alone spatial autocorrelation. Please explain or rephrase. - lines 51-56 "Another shortcoming relating to EDF and RFsp is that..." As EDF and RFsp did not intend to keep the number of coordinate covariates variable I would suggest "reduced flexibility" instead of "shortcoming". - line 65-66. "...it should be possible to optimise it" Please be specific: is it possible or not? Does it make sense to optimise it? Why did the authors then merely test all numbers of coordinate covariates?

Further comments:

- Please delete equations (1) – (3). This is simple trigonometry. Please also consider adapting the symbology: b2 is the knew oblique coordinate that replaces b1 (=x) and a1 (=y) not only b1 as somehow suggested by naming it b2. - lines 135-136. Please add the tested mtry values - line 136. Please explain how extratrees allows for sub-optimal splits - Does the approach work on any type of coordinate system? I suppose coordinates have to be projected?

---

## Author Comment (AC2) · 28 Feb 2020

**Reply to Referee #1 on our manuscript "Oblique geographic coordinates as covariates for digital soil mapping"**

5    We thank Referee #1 for the well-thought and qualified comments on our manuscript. In the following, we will address the referee's comments and describe the changes that they have occasioned in the manuscript.

COMMENT

The present paper is well written and structured. Moreover, the study aims to make a
10    contribution to the field of DSM by providing a novel methodological framework based on the usage of coordinates. This is something that can be considered as rather 'out of the box thinking', because most attention in the international literature goes either to the use of advanced geostatistical methods (e.g. capturing the spatial autocorrelation through kriging) or external drift fitting based on '(environmental) co-variates' or a combination of both. Hence,
15    the work certainly merit respect for its originality and the methodological framework seems to provide useful thoughts to be considered in future DSM-studies. However, I also see some shortcomings which should be addressed/considered in order to maximize its potential to be applied widely, and as such, I am looking forward receiving the authors replies on the associated comments and suggestions presented below.

20    REPLY

We thank the referee for seeing the value in our research. We will address the shortcomings listed by the referee in the following.

COMMENT

Major Comments: I believe that the main issue with this research is that it considers only one
25    rather small field characterized by 1 remarkable / specific spatial structure as regards the variation of SOM (i.e. one spot/area with clearly higher values) in order to test the validity of the present new methodology, whereas the authors claim that the method will be highly useful for mapping soil properties in larger areas. Hence, I believe that the present methodology requires further testing by considering larger areas (e.g. catchment-regional
30    scale) with more complex spatial patterns in SOM in order to prove the validity of the statements that have been made in this respect. Moreover, it would also be interesting to consider other key soil variables (besides SOM) to check whether the usages of oblique geographic coordinates as covariates could be seen as a universal DSM approach. In this context, I believe that using a national soil inventory database could be a good way forward. I
35    may understand that this might not be possible in this study, but I still believe that this should

be mentioned clearly (as a critical note) in the discussions (and maybe be picked up by the authors in future research).

REPLY

40 We agree with the referee that it is a shortcoming that we only test the method for one soil property in one area. We believe that it is a vital next step to test the method in other areas and for other soil properties. We already mention this issue in the conclusion (L323 – L325). However, we see that we could add further emphasis on the subject.

CHANGES

In order to increase emphasis on the necessity of testing OGC on more datasets, we will
45 rephrase L323 – L325:

"One of the main shortcomings of this study is the fact that we have only tested OGC and compared it to other methods for one soil property in one agricultural field. This means that our study could not fully assess the relative usefulness of the method. A vital next step is therefore to test OGC on a range of soil properties in a larger area. It will especially be
50 relevant to test it for large areas, as the spatial patterns of soil properties in large areas are typically more complex."

COMMENT

When I have a look at the performance of the different mapping methods (as presented in the Violin plots in figure 7), it seems to me that your new OGC (+AUX) method only results in
55 (very) small improvements as compared to some other (more commonly used) methods such as Kriging. Hence, I was wondering whether this improvement is statistically significant? And if this might still be the case when either (i) another field (characterized by a different spatial pattern), (ii) another soil variable or (iii) larger geographical extent are considered?

REPLY

60 As we state in L149 – L153 and the caption for Table 2, some of the differences in accuracy are statistically significant and some are not. We used the same 100 repeated training/test splits for all methods, and this allowed us to carry out pairwise t-tests between the accuracies of the methods. We then ranked the methods using the results of these t-tests. Methods that did not have statistically significant differences for a given metric received the same rank for
65 that metric, but methods with statistically significant accuracies received different ranks. For example, OGC + AUX and RFsp + AUX always received the same rank, as there were no statistically significant differences in the accuracies. Meanwhile, kriging always received a higher rank than AUX, because the differences in their accuracies were statistically significant.

70 Kriging, RFsp + AUX and OGC + AUX all received the highest rank for two out of three accuracy metrics. We therefore regard these three methods as equally accurate. We already

state in the manuscript that we regard these three methods as most accurate (L210). However, we see that we have not explicitly stated that we regard them as equally accurate.

Furthermore, we cannot know the results of t-test carried out for accuracies in other/larger areas and for other soil properties, as these results currently do not exist.

CHANGES

We will rephrase L210 to explicitly state that we regard the three methods as equally accurate:

"We therefore regard these three methods as the most accurate methods. Furthermore, we regard these three methods as equally accurate, as none of them were universally more accurate than the other two methods."

COMMENT

Minor Comments: I'm not too sure if it is entirely appropriate to use R2 as a measure to compare the different methods, because (i) a very high R2 value may also mean an 'overfit' and (ii) each method has it own degree of (model) complexity. Hence, I guess that it could be a good idea to take (also) another statistical measure into consideration that specifically aims to evaluate the methods' performance taking into account its complexity (in order to avoid overfitting)?

REPLY

We understand the referee's concern, as a very high $R^2$ on a training dataset can indicate overfitting of a model. However, we report $R^2$ for 25% holdout datasets not used in the models. Our $R^2$ values therefore indicate the predictive capabilities of the models rather than their fit on the training data. Furthermore, we are not aware of any measures of accuracy that account for complexity in Random Forest models. We are even less aware of any accuracy measures capable of comparing complexities of conceptually very different models, such as Random Forest and kriging. We think most readers will be aware that kriging is much simpler method than Random Forest. In fact, we explicitly state this in the manuscript (L295 – L296).

COMMENT

Figure 1 - Subpanel C: Showing hill shade is not enough to give the reader an insight into the topographical configuration of the field. Hence, I suggest adding contour lines.

REPLY

We thank the referee for this helpful comment. We agree that adding contour lines improves the visualization of the topography of the study area.

CHANGES

Due to the referee's comment, we have prepared a new version of Figure 1, where we have added 2 m contour lines. We will include this updated figure in the final version of the manuscript:

---

## Author Comment (AC3) · 28 Feb 2020

**Reply to Referee #2 on our manuscript "Oblique geographic coordinates as covariates for digital soil mapping"**

We thank the referee for the qualified and insightful comments on our manuscript. In the following, we will address the referee's comments and describe the changes that we have made to the manuscript because of the comments.

COMMENT

The manuscript "Oblique geographic coordinates as covariates for digital soil mapping" from Møller et al. presents a valuable contribution to integrate predictor information on spatial position into machine learning approaches for digital soil mapping. It, thereby, seeks to overcome the known problem of orthogonal artefacts sometimes introduced by the usage of xy-coordinates as covariates in recursive partitioning algorithms. While commonly applied covariates usually relate to site characteristics that approximate the soil forming factors, the inclusion of coordinates provides a chance to reflect further spatial patterns we are not necessarily aware of. The authors show that the usage of a multitude of oblique spatial coordinates reflects spatial anisotropy. Major spatial axes identified through predictor importance measures may then give a hint on the geographic direction of the underlying processes as the authors demonstrate. The article compares the new approach (OGC) to existing approaches such as Euclidean distance fields (EDF) and spatial Random Forest (RFsp). The article is written using adequate language and it follows a clear structure. The figures are well prepared. Furthermore, it is a rare, but highly welcome choice of the authors to provide the R code of their approach. While the authors very clearly demonstrate the power of their approach particularly due to the clear figures and the comparison to similar approaches, certain aspects would require reconsideration:

REPLY

We thank the referee for the support for our manuscript. Furthermore, we are happy that the referee appreciates our choice to share the code for our study. We will consider the issues that the referee raises in the following.

COMMENT

- I do not understand why the OGC+AUX approach is not directly compared to regression kriging, but to ordinary kriging. Ordinary kriging would require a stationary mean which is not given in this particular research setting. Accordingly, a regression model would first have to be fitted to model the trend from covariate data, while then spatial autocorrelation in the residuals will be accounted for by ordinary kriging of the residuals. While the regression

35    model is fitted by random forest, this would also allow for direct comparability. The authors provide rather vague arguments against regression kriging (lines 27-32).

REPLY

Our study focuses on one-step methods, as one of the goals in developing OGC is to create a feasible one-step method. Two-step approaches such as regression-kriging require researchers

40    to interpret two models at once, which can confound analyses of uncertainty and the processes that govern the spatial distribution of soil properties. We believe that this is a relevant consideration, but it is not our main reason for omitting regression-kriging. Our first reason for this choice is that a previous study carried out in the same area showed that kriging predicted SOM more accurately than regression-kriging using both Cubist and Random

45    Forest models (Pouladi et al., 2019). When a relatively simple method outperforms complex approaches, we believe that it is right to consider the complex approaches as redundant. Without this previous finding, we believe that it would have been relevant to include regression-kriging in the comparison.

CHANGES

50    We see that the manuscript does not clearly state our reasons for omitting regression-kriging. We will therefore add the following paragraph to section 2.3:

"A previous study in the same area showed that kriging predicted SOM more accurately than regression-kriging (Pouladi et al., 2019). We therefore omitted regression-kriging from the analysis, although, without this previous finding, it would have been relevant to include it."

55    COMMENT

The data in this study display spatial autocorrelation. Specifically, a range of 139 m is mentioned. This is not surprising due to the high spatial data density. Furthermore, the authors mention a couple of processes that may have caused this spatial dependency. However, this aspect is not accounted for in the evaluation approach. 100 random splits 75/25

60    (training/ test set) make it very likely that spatially autocorrelated sampling points will end up in the test and training set for the majority of the 100 splits. As a consequence, the test sets are not independent of the training sets and will lead to overly optimistic error values. This aspect at least needs to be mentioned. Particularly, in the context of Figure 7. The argumentation line of the introduction requires some improvement.

65    REPLY

Our main priority in the study is to compare the accuracies of several methods, not to assess their accuracies in absolute terms. Furthermore, we do not consider the issue of spatial autocorrelation to be as grievous as to warrant attention in the manuscript. Firstly, geostatistical approaches such as kriging would be useless if there was no spatial

70    autocorrelation in the data. Secondly, the sample distribution in the field is very even, and as a result, only very few areas in the field are more than 20 meters from the nearest sample, and all areas are within the range of spatial autocorrelation. Therefore, having training and test

samples within the range of spatial autocorrelation actually represents the general conditions in the field quite well. We therefore do not believe that our accuracy metrics are very much overly optimistic. If we were to extrapolate our results to a larger area, spatial autocorrelation would be an issue to consider, but this is not the goal of our study.

75

COMMENT

Certain aspects need to be better clarified:

- The main advantage of OGC+AUX over using only XY+AUX is the high number of coordinates, as the usage of only two oblique coordinates would lead to similar artefacts as demonstrated in the results. - The usage of coordinates as predictors in a regression model differs from fitting a geostatistical model to the residuals of a regression model. The approach closest to fitting a semivariogram is RFsp, as it accounts for the distance between points. However, it comes at the cost of introducing a high number of covariates as the authors state, correctly. It is important that the authors also compare their approach to RFsp, but the difference in calculating a different set of coordinates and taking the distance between points into account should be explicitly mentioned. In contrast, OGC+AUX and EDF+AUX really follow a similar approach in calculating a set of different coordinates. OGC+AUX is demonstrated to be superior to EDF+AUX. - Overall, whether it is worse to make the effort of calculating a high number of oblique coordinates could only be decided while being compared to regression kriging.

80

85

90

REPLY

We agree with the referee, and we see the need for further clarification. We will add several statements to the final version of the manuscript for this purpose.

95    CHANGES

We will add the following statements:

L44: "The main advantage of this approach [RFsp] is that it incorporates distances between observations in a similar manner to geostatistical models".

L301: "Of the previous approaches, OGC is most similar to EDF, as it used the x- and y-coordinates, and the distances to the corners of the study area resemble coordinates. On the other hand, RFsp is more similar to geostatistical models, as it relies on distances between observations. However, this similarity comes at the cost of calculating a large number of distance rasters."

100

L319: "The method [OGC] eliminated the orthogonal artefacts that arise from use of x- and y-coordinates and also achieved higher accuracies than maps created with only two coordinate rasters."

105

COMMENT

There are a couple of statements that are problematic. Please consider rephrasing:

- lines 19-21 "…decision tree algorithms…are immune to correlated and redundant covariates". There are a couple of publications that show the contrary.

REPLY

Our experience has shown that decision trees are less vulnerable to correlated and redundant covariates than other model types, such as artificial neural networks. However, we admit that this does not constitute a full immunity.

CHANGES

We see that our statement is not correct, and we will therefore remove it from the final version of the manuscript.

COMMENT

- line 29 "By kriging the residuals…soil mappers have been able to reduce or remove spatial bias". We usually fit a geostatistical model to explain spatial autocorrelation not to remove spatial bias. Please also correct throughout the manuscript, e.g. lines 45/46.

REPLY

We agree with the referee that our phrasing is incorrect, and we will therefore change it (see below). However, the phrasing in lines 45 – 46 is in line with the study to which we refer. We quote the authors: "Further analysis shows that in both cases there is no remaining spatial autocorrelation in the residuals […]. Hence, both methods have fully accounted for the spatial structure in the data" (Hengl et al., 2018). The authors of this study refer to a figure, which shows a pure nugget variogram for the residuals of their model.

CHANGES

We will rephrase the sentence in question:

"By kriging the residuals of the predictive model and adding the kriged residuals to the prediction surface, soil mappers have been able to explain spatial autocorrelation and achieve higher accuracies."

COMMENT

- line 47 "...methods are able to integrate spatial relationships…" I am not convinced that by the mere consideration of coordinates we account for spatial relationships, leave alone spatial autocorrelation. Please explain or rephrase.

REPLY

We agree that our use of the term "spatial relationships" is inaccurate.

CHANGES

We will replace the term "spatial relationships" with the term "spatial trends" throughout the manuscript.

COMMENT

- lines 51-56 "Another shortcoming relating to EDF and RFsp is that…" As EDF and RFsp did not intend to keep the number of coordinate covariates variable I would suggest "reduced flexibility" instead of "shortcoming".

REPLY

We agree with the referee, and we will rephrase as requested.

CHANGES

We will rephrase L51:

"EDF and RFsp also have limited flexibility as both methods specify the number of geographic data layers a priori."

COMMENT

- line 65-66. "…it should be possible to optimise it" Please be specific: is it possible or not? Does it make sense to optimise it? Why did the authors then merely test all numbers of coordinate covariates?

REPLY

We see that the sentence is not very clear. We will therefore rephrase it.

CHANGES

We will rephrase lines 65 – 66:

"Furthermore, the number of oblique angles is adjustable, and soil mappers can therefore choose a number that suits their purpose. Some mapping tasks may require a higher number of oblique angles than others, and soil mappers can therefore increase the number as necessary. Alternatively, if a small number of oblique angles suffices, soil mappers can reduce their number and thereby shorten computation times."

COMMENT

Further comments:

- Please delete equations (1) – (3). This is simple trigonometry.

REPLY

We agree with the referee.

CHANGES

We will delete equations 1 – 3.

COMMENT

Please also consider adapting the symbology: b2 is the knew oblique coordinate that replaces
b1 (=x) and a1 (=y) not only b1 as somehow suggested by naming it b2.

REPLY

Our reason for naming $b_2$ is that it forms one of the sides of the right triangle $a_2b_2c$. We will
therefore not rename it, as it would obscure interpretation of Figure 2. However, we see that
the equations and the figure do not sufficiently stress the fact that the length of $b_2$ is equal to
the new oblique coordinate.

CHANGES

We will add "OGC" to equation 4, to stress that OGC is equal to the length of $b_2$:

$$OGC = b_2 = \sqrt{a_1{}^2 + b_1{}^2} * \cos\left(\theta - \tan^{-1}\frac{a_1}{b_1}\right)$$

COMMENT

- lines 135-136. Please add the tested mtry values

REPLY

In each model, we tested five *mtry* values at even intervals between 2 and *NC*, where *NC* is
the total number of covariates (counting both auxiliary data and spatially explicit covariates).
The tested *mtry* values therefore depended on the method, and the number of covariates
differed between methods.

CHANGES

We will add this explanation to the paragraph, starting at line 137:

"We tested *mtry* values at even intervals between 2 and the total number of covariates,
including auxiliary data and spatially explicit covariates. The tested *mtry* values therefore
varied depending on the number of covariates."

COMMENT

- line 136. Please explain how extratrees allows for suboptimal splits

REPLY

We will rephrase the sentence to better clarify how *extratrees* works.

CHANGES

We will rephrase the sentence as follows:

"The *extratrees* splitting rule generates random splits, as opposed to the *variance* splitting rule, which chooses optimal splits. Per default, *extratrees* generates one random split for each covariate and then chooses the random split that gives the largest variance reduction (Geurts et al., 2006). It therefore leads to a greater degree of randomization."

COMMENT

- Does the approach work on any type of coordinate system? I suppose coordinates have to be projected?

REPLY

This is a very interesting question, which we have given some though, although we have not included these thoughts in the first version of the manuscript. In the study, we use UTM coordinates, which have the advantage that the x- and y-coordinates have the same unit. Furthermore, it is reasonable to treat relatively small study areas as two-dimensional planes. In practical terms, OGC may also work reasonably well for larger areas with other coordinate systems, such as latitude/longitude systems. However, interpretation would not be as straightforward as in this study.

Using OGC at a global extent would probably require changes to the method. Because longitude is circular, points located on different sides of 180° L would have drastically different coordinates, even if the actual distances between them were short. One solution to this problem could be to replace the present version of OGC with latitudes rotated at various angles around a pair of equatorial axes. However, the implementation and testing of such an approach is far outside the scope of this study.

Due to the interest of this question, we will shortly address it in the conclusions section of the revised manuscript.

CHANGES

We will add the following statement to the conclusions section:

"One should note that we carried out this study for a small area using UTM coordinates as input. Using OGC for larger areas and other coordinate systems may require alterations to the method."

**References**

Geurts, P., Ernst, D. and Wehenkel, L. Extremely randomized trees. Mach. Learn. 63(1), 3-42. http://dx.doi.org/10.1007/s10994-006-6226-1, 2006.

Hengl, T., Nussbaum, M., Wright, M.N., Heuvelink, G.B. and Gräler, B. Random forest as a generic framework for predictive modeling of spatial and spatio-temporal variables. PeerJ 6, e5518. http://dx.doi.org/10.7717/peerj.5518, 2018.

Pouladi, N., Møller, A.B., Tabatabai, S. and Greve, M.H. Mapping soil organic matter contents at field level with Cubist, Random Forest and kriging. Geoderma 342, 85-92. http://dx.doi.org/10.1016/j.geoderma.2019.02.019, 2019.